



# Ensemble-based assimilation of fractional snow covered area satellite retrievals to estimate snow distribution at a high Arctic site

Kristoffer Aalstad[1], Sebastian Westermann[1], Thomas Vikhamar Schuler[1], Julia Boike[2], and Laurent Bertino[3]

[1]Department of Geosciences, University of Oslo, P.O. Box 1047, Blindern, 0316 Oslo, Norway
[2]Alfred Wegener Institute Helmholtz Center for Polar and Marine Research, Telegrafenberg A43, 14473 Potsdam, Germany
[3]Nansen Environmental and Remote Sensing Center, Thormøhlensgate 47, Bergen 5006, Norway

*Correspondence to:* Kristoffer Aalstad (kristoffer.aalstad@geo.uio.no)

**Abstract.** Snow, with high albedo, low thermal conductivity and large water storing capacity strongly modulates the surface energy and water balance, thus making it a critical factor in mid to high-latitude and mountain environments. At the same time, already at medium spatial resolutions of $1\,\mathrm{km}$, estimating the average and subgrid variability of the snow water equivalent (SWE) is challenging in remote sensing applications. In this study, we present an ensemble-based data assimilation scheme

that assimilates fractional snow covered area (fSCA) retrievals into a simple snow model forced by statistically downscaled reanalysis data so as to estimate peak SWE distributions at the kilometer scale. The basic idea is to relate the timing of the snow cover depletion (that is accessible from satellite products) to pre-melt SWE, while at the same time obtaining the subgrid scale distribution. Subgrid SWE is assumed to be lognormally distributed, which can be translated into a modeled time series of fSCA by means of the snow model. Assimilation of satellite-derived fSCA hence facilitates the constrained estimation of the

average SWE and coefficient of variation, while taking into account uncertainties in both the model and the assimilated data sets. As an extension to previous studies, our method makes use of the novel (to snow data assimilation) ensemble smoother with multiple data assimilation (ES-MDA) scheme combined with analytical Gaussian anamorphosis to assimilate time series of MODIS and Sentinel-2 fSCA retrievals. The scheme is applied to high-Arctic sites near Ny Ålesund (79°N, Svalbard, Norway) where in-situ observations of fSCA and SWE distributions are available. The method is able to successfully recover

accurate estimates of peak subgrid SWE distributions on most of the occasions considered. Through the ES-MDA assimilation, the root mean squared error (RMSE) for the fSCA, peak mean SWE and subgrid coefficient of variation is improved by around 75%, 60% and 20% respectively when compared to the prior, yielding RMSEs of 0.01, $0.09\,\mathrm{m\ water\ equivalent\ (w.e.)}$ and 0.13 respectively. By comparing the performance of the ES-MDA to that of other ensemble-based batch smoother schemes, it was found that the ES-MDA either outperforms or at least nearly matches the performance of the other schemes with

regards to various evaluation metrics. Given the modularity of the method, it could prove valuable for a range of satellite-era hydrometeorological reanalyses.

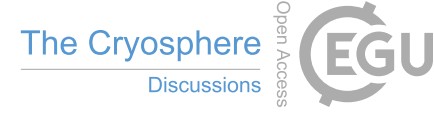

# 1   Introduction

The seasonal snow cover exhibits a high albedo, low thermal conductivity, large water holding capacity, strong smoothing effect on the terrain as well as a considerable variability both in depth and horizontal extent. Consequently, the distribution of the seasonal snow cover in time and space is a key control on the terrestrial surface energy and mass balance (e.g. Boike et al., 2003). Thereby, snow is a central modulator of the global radiation balance and hydrological cycle and is thus one of the drivers of the atmospheric circulation and the associated climate (Liston, 1999; Andreadis and Lettenmaier, 2006). For example, in mid and high latitude regions, as well as in mountainous areas, (a combined region that accounts for about one quarter of the global gross domestic product) the snow melt plays a dominant role in the seasonal patterns of the stream flow (Barnett et al., 2005). In terms of water resource management, there is thus much to be gained in being able to map the spatiotemporal distribution of snow water equivalent (SWE) across this extensive region.

The primary controls on the distribution of SWE are topography, vegetation, precipitation, wind, radiation and avalanching (Sturm and Wagner, 2010). Of these the former two are relatively fixed temporally. The remaining dynamic controls vary significantly not just in space, but also in time. All of these controls occur across a range of scales and thus cause great variability in snow depth. For example, snow tends to be affected by wind drift in non-forested regions (e.g. Gisnås et al., 2014) leading to accumulation in areas with preferential deposition such as topographic depressions or the lee-side of a ridge, but the scale of such features varies dramatically across the landscape (Clark et al., 2011). Therefore, especially due to the dynamic controls, the mapping of SWE distributions is particularly challenging. Nonetheless, the processes occurring at a given site are sometimes quite consistent from year to year and so the SWE distribution pattern in any given year may be quite similar to the climatological snow distribution pattern (Sturm and Wagner, 2010). For mapping SWE distributions over large areas, manual measurement surveys usually have both limited support, large spacing and small extent (Blöschl, 1999) and are thus exhausting, impractical and expensive as well as potentially hazardous in steep terrain due to avalanches. Instead, modeling, remote sensing or a combination of the two can be employed to map SWE.

Snow models range in complexity from relatively simple single layer models, such as the Utah Energy Balance model (UEB; Tarboton and Luce, 1996; You et al., 2014), to detailed multi-layer snowpack models such as Crocus (Vionnet et al., 2012) and SNOWPACK (Bartelt and Lehning, 2002). Many snow models (e.g. ALPINE3D; Lehning et al., 2006) can also be run in distributed mode so as to predict the snow distribution over large areas. The accuracy of the model results are limited by the hydrometeorological forcing data, be it from reanalyses or local measurements, whose errors are typically the major source of uncertainty in snow modeling (De Lannoy et al., 2010; Raleigh et al., 2015). A problem with snow models is that they are often developed as point scale models and even if they are run as distributed models there is no guarantee that the grid scale values predicted by the model are representative of the corresponding process scale (Blöschl, 1999). Indeed, for the example of a model forced by near-point scale hydrometeorological measurements, the model predictions will only be representative of some grid-scale if that particular point is representative of the mean conditions within the grid cell which is generally unlikely





given the large spatial variability in snow depths (Clark et al., 2011). To circumvent this problem probabilistic snow depletion curve (SDC) parametrizations have been developed (Liston, 1999; Luce and Tarboton, 2004; Liston, 2004). Therein, a distribution function is assigned to the SWE within a grid cell at peak accumulation and, along with an assumption of uniform melt across the grid cell, this allows for a direct relationship between the mean SWE and the fractional snow covered area (fSCA) of

the grid cell. Liston (2004) used such an SDC parametrization in conjunction with a dichotomous key (a simple classification scheme) for the subgrid coefficient of variation of SWE with the ClimRAMS model at 80 km resolution to map the fSCA over North America; including the SDC in ClimRAMS increased the total snow-covered area considerably compared to the control run. More recently, Aas et al. (2017) used a mosaic, or tiling, approach to represent subgrid snow variability in the weather research and forecasting (WRF) model coupled to the Noah land surface scheme at 3 km resolution over southern Norway. The

tiling simultaneously reduced the cold bias in the model and greatly improved the match to the observed fSCA evolution. Still, as mentioned, modeling alone is in itself arguably not an effective tool for mapping SWE distributions in hindcast mode due to the inherently large uncertainties in the hydrometeorological forcing. Instead, models need to be combined with relevant data from remote sensing.

Snow-related data sets can be acquired from a variety of remote sensing platforms. The Gravity Recovery and Climate Experiment (GRACE) twin satellites allow for the retrieval of terrestrial water storage (TWS), from which SWE can be recovered at around 100 km spatial resolution (e.g. Niu et al., 2007). Similarly passive microwave (PM) satellite sensors can retrieve SWE based on brightness temperature at a somewhat finer spatial resolution of around 25 km. However, PM SWE retrievals have problems over forested areas and complex topography, as well as for wet and deep snowpacks (Foster et al., 2005). Both

gravimetric and PM sensors are able to retrieve SWE independent of cloud coverage, generally resulting in gap-free time series. While not capable of measuring SWE, moderate resolution optical sensors such as MODIS can retrieve binary information on snow cover (i.e. snow or no snow), fSCA and snow grain size (Hall et al., 2002; Salomonson and Appel, 2004, 2006; Painter et al., 2009) at approximately 500 m spatial resolution at a daily revisit frequency. In addition, there exists high resolution optical sensors, e.g. on board the LandSat and Sentinel-2 satellites, that can map snow cover and fSCA at 30 m spatial resolution

(e.g. Cortés et al., 2014). Such optical sensors can not see through a cloud cover, which results in extended data gaps in most snow-covered regions. To obtain continuous time series, it is necessary to either interpolate such remote sensing data in time and space, or ingest them into models.

Data assimilation (DA) methods can objectively fuse uncertain information from observations and models. The simplest

form of snow data assimilation are the so-called deterministic SWE reconstruction techniques (Girotto et al., 2014b) that make use of direct insertion of remotely sensed fSCA data. Such schemes back-calculate peak SWE from the disappearance date of the snow cover (as determined from fSCA retrievals) using snow melt models. Martinec and Rango (1981) used retrievals of fSCA from LandSat images during the melt season in conjunction with a simple degree day snow melt model to estimate the peak mean SWE at 1 km spatial resolution. Similarly, Cline et al. (1998) also used fSCA retrievals based on LandSat imagery,

but this time combined with a distributed energy balance model, to reconstruct the SWE distribution at 30 m resolution. More



recently, Molotch and Margulis (2008) used fSCA information from multiple sensors (LandSat ETM+, MODIS and AVHRR) to reconstruct the SWE distribution at 100 m resolution. Durand et al. (2008) introduced a probabilistic framework for SWE reconstruction which was based on assimilating synthetic fSCA retrievals during the ablation into the SSiB3 model coupled to the probabilistic SDC of Liston (2004) using the ensemble smoother (ES; Van Leeuwen and Evensen, 1996) in batch mode

(c.f. Dunne and Entekhabi, 2005). The assimilation of fSCA in this synthetic, or 'twin', experiment was used to correct annual biases in the snowfall, and enabled the recovery of the SWE distribution at 90 m resolution. Through the assimilation the bias and root mean square error (RMSE) of the SWE estimation was reduced by 86% and 78% respectively for pixels with less than 90% vegetation cover. Girotto et al. (2014b) discussed how the method of Durand et al. (2008) was a generalization of the previous deterministic SWE reconstruction techniques cast in a probabilistic framework. In addition, Girotto et al. (2014b)

assimilated LandSat based fSCA retrievals with the same DA framework to recover the peak SWE distribution at 90 m resolution. A significant reduction in RMSE was found when comparing the probabilistic to the deterministic SWE reconstruction techniques. Subsequently, Girotto et al. (2014a) used the same framework to perform a 27 year reanalysis of SWE distributions at 90 m resolution. Recently, Margulis et al. (2015) modified the probabilistic SWE reconstruction approach by adopting a particle batch smoother (PBS) as opposed to the ES for the assimilation of fSCA retrievals to estimate the SWE distribution at

90 m resolution. The PBS was found to perform favorably compared to the ES, considerably reducing the RMSEs compared to in-situ measurements. Based on this work, Margulis et al. (2016) adopted the same framework to construct a 30 year reanalysis of SWE over the Sierra Nevada (USA) using LandSat-based fSCA retrievals also at 90 m resolution. The resulting posterior SWE estimates had an RMSE of 0.11 m and 0.13 m and correlation coefficient of 0.97 and 0.95 for snow pillows and snow courses respectively.

In addition to deterministic and probabilistic SWE reconstruction, there have been several other snow data assimilation techniques employed in recent studies. Andreadis and Lettenmaier (2006) assimilated MODIS fSCA retrievals into the VIC model through the ensemble Kalman filter (EnKF; Evensen, 1994) using a simple SDC for the SWE-fSCA inversion. However, the improvement, compared to the open-loop (no DA) run, in SWE estimates was only modest, which was also found

in similar studies using the EnKF (Clark et al., 2006; Slater and Clark, 2006). A fully Bayesian DA technique was used by Kolberg and Gottschalk (2006) to assimilate fSCA retrievals from LandSat 7 ETM+ images into a snow model with a probabilistic SDC to estimate the peak SWE distribution at 1 km resolution. They found a significant reduction in uncertainty when two retrievals were assimilated simultaneously as opposed to sequentially. At the continental scale, a multisensor assimilation of both GRACE TWS and MODIS fSCA using the ES and EnKF for TWS and fSCA respectively yielded significant

improvements compared to the open loop (Su et al., 2010). De Lannoy et al. (2010) used the EnKF in a twin experiment to assimilate synthetic 25 km scale passive microwave SWE retrievals into a 1 km resolution land surface model, resulting in up to a 60% improvement in RMSE compared to the open-loop. The approach was extended to a real and multisensor experiment by jointly assimilating AMSR-E SWE and MODIS fSCA retrievals (De Lannoy et al., 2012). At sites with shallow snow-packs the combined SWE and fSCA assimilation yielded a marked improvement in the RMSE compared to the open loop, the



improvement was marginal in areas with deeper snowpacks given the limitations of PM SWE retrievals (c.f. Foster et al., 2005).

Of late, particle filter (PF; see Van Leeuwen, 2009) schemes have been gaining popularity in snow DA studies (Charrois et al., 2016; Magnusson et al., 2017). For example, Charrois et al. (2016) assimilated synthetic MODIS-like optical reflectance retrievals into the detailed snowpack model Crocus using the sequential importance re-sampling PF at a point scale, which resulted in a 45% reduction in RMSE in both snow depth and SWE when compared to the open loop. It is worth emphasizing that the most popular schemes in the snow DA community, i.e. both the EnKF and the PF, are filters meaning that they are sequential techniques. As such, they are Markovian of order 1 ('memoryless'); i.e. the future state at a given point in time depends only on the present state. Moreover, observations are assimilated sequentially with only the current observation affecting the current state. Batch smoothers (see Dunne and Entekhabi, 2005), on the other hand, take into account the entire history of a model trajectory within a batch (observation window), and as such have memory (non-Markovian), so that they are better suited for reconstruction, i.e. reanalysis, type problems. Within petroleum reservoir geophysics, an entire subfield is dedicated to history matching (c.f. Le et al., 2016) that makes ubiquitous use of ensemble batch smoother type schemes.

In this study, we build on the probabilistic SWE reconstruction techniques outlined in Girotto et al. (2014b) and Margulis et al. (2015) to recover subgrid SWE distributions (SSD) for a study area in the high Arctic, based on fSCA retrievals from MODIS and Sentinel-2. The novelty of our study lies in the use of an iterative batch smoother scheme, namely the ensemble smoother with multiple data assimilation (ES-MDA; Emerick and Reynolds, 2013), which is popular in the aforementioned history matching community. To update parameters that are bounded in physical space we also make explicit use of analytical Gaussian anamorphosis (Bertino et al., 2003). We investigate the performance of the ES-MDA in terms of SWE reconstruction and compare it to the ES and the PBS employed by Girotto et al. (2014b) and Margulis et al. (2015) respectively. The results are evaluated against independent in-situ measurements of fSCA and peak SSD conducted over six snow seasons. As both the forcing data and satellite retrievals are available globally, the presented modular technique holds great potential for estimating peak SSD at a range of spatial scales.

## 2 Study area

### 2.1 Physical characteristics and climate

The high-Arctic study area is located in NW Svalbard close to the research town of Ny-Ålesund ($78°55'$N, $11°50'$E) on the Brøgger peninsula, where in-situ measurements are available from three sites (Figure 1). "Bayelva" about two kilometers west of Ny-Ålesund is the main study site from where multi-year in-situ records on e.g. surface energy balance, permafrost thermal regime and snow distribution are available (Boike et al., 2003; Westermann et al., 2009; Gisnås et al., 2014). In addition, snow surveys for a single season are available from "Steinflåen plateau" and "Kvadehuksletta". All sites feature undulating topography, with surfaces characterised by patterned gorund features, leading to strong differences in snow cover due to wind drift. While both Bayelva and Kvadehuksletta are located between 10 and 50 m a.s.l., the Steinflåen plateau is at a slightly higher





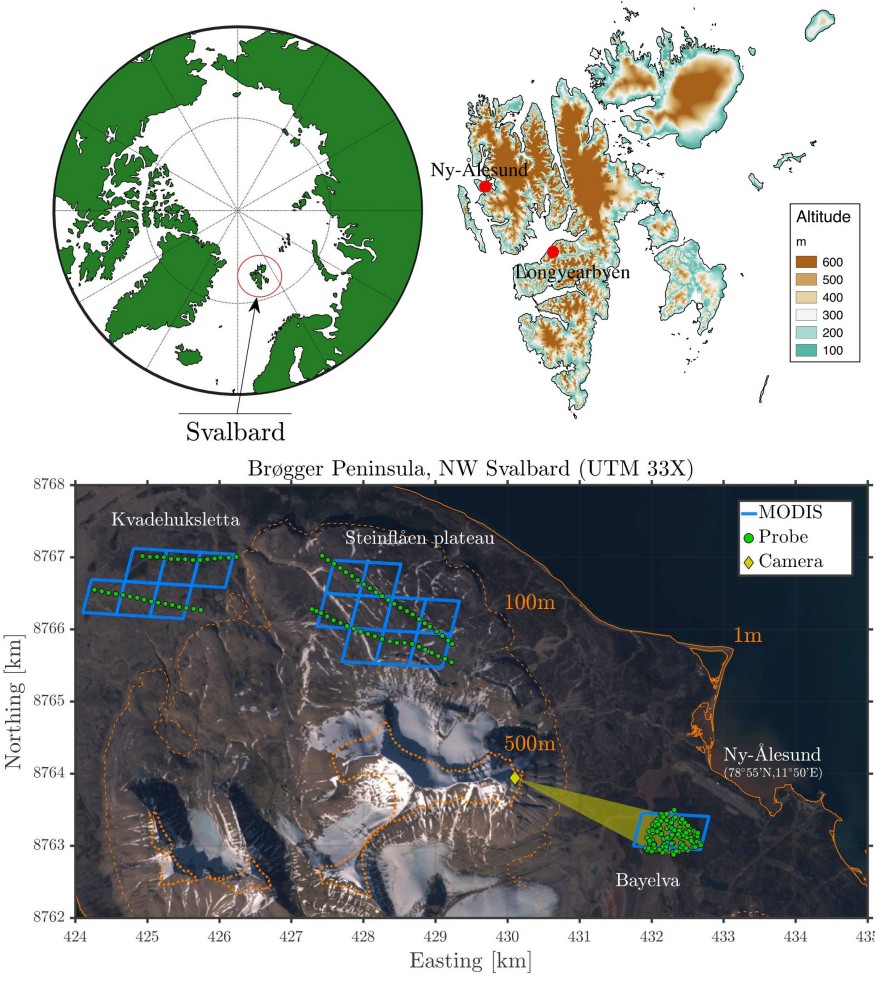

**Figure 1.** Top left: location of the Svalbard archipelago (in red) in the Arctic. Top right: map of Svalbard, the location of the study area is close to Ny-Ålesund. Bottom: Sentinel-2A true color image (taken 02.07.2016) of the western Brøgger Peninsula with the three study sites Kvadehuksletta, Steinflåen plateau and Bayelva (from west to east); green dots: snow survey probe locations; blue polygons: MODIS pixels; yellow diamond: automatic camera system on Scheteligfjellet; yellow shading: field of view of the camera; contour lines courtesy of the Norwegian Polar Institute (2014) DEM.

elevation of around 200 m. Furthermore, the wind regime features considerable differences between the sites, with Kvadehuksletta exposed to most wind directions, while Bayelva and Steinflåen plateau are partly sheltered by mountain ranges. The study sites are located within the continuous permafrost zone (Boike et al., 2003) with an active layer depth on the order of 1.5 m at the Bayelva site (Westermann et al., 2009).





The Bayelva site is located around the heavily instrumented Bayelva climate and soil monitoring station (see Westermann et al., 2009). The surrounding area has been the subject of extensive study on a wide range of topics spanning permafrost (Roth and Boike, 2001; Boike et al., 2008; Westermann et al., 2011a), the surface energy balance (Boike et al., 2003; Westermann et al., 2009), $CO_2$ exchange (Lüers et al., 2014; Cannone et al., 2016), ecology (Kohler and Aanes, 2004), snow (Gisnås et al., 2014; López-Moreno et al., 2016), hydrology (Nowak and Hodson, 2013) and satellite retrieval validation (Westermann et al., 2011b, 2012). At Bayelva, the surface cover alternates between bare soil, rocks and sparse vegetation in the form of low vascular plants, mosses and lichens (Westermann et al., 2009; Gisnås et al., 2014). The surface cover at Kvadehuksletta is similar to that at Bayelva, but the more elevated Steinflåen plateau is predominantly covered by loose rocks.

The climate of western Svalbard is influenced by the relatively warm West Spitsbergen current causing a maritime climate with mild winters and cool summers for this latitude (Esau et al., 2012). At Ny-Ålesund the winter, summer and annual (1981-2010) average air temperatures were -12.0 °C, 3.8 °C and -5.2 °C, respectively, while the average annual precipitation was 427 mm (Førland et al., 2011). Between September/October and May most of the precipitation falls as snow, although rain on snow events have become more frequent due to the recent observed warming of the local climate (Nowak and Hodson, 2013; López-Moreno et al., 2016). The perennial snow cover usually forms in late September or early October and lasts until mid June to early July, with around a month long melt season (Winther et al., 2002). Measurements at the Bayelva site show that the energy available for snow melt is dominated by radiation, both shortwave and longwave (Boike et al., 2003; Westermann et al., 2009). In addition, the energy required to warm the frozen ground beneath the snow needs to be considered in the energy balance during the snow melt (Boike et al., 2003).

## 2.2 Field measurements

Manual surveys of snow depth and density were carried out in late April/early May for six years at the Bayelva site (see Gisnås et al., 2014, for details) and for one year (2016) at the two other sites (Table 1). At this time, just before the onset of the melt, the snow depth is near maximum but the snowpack is still dry. The snow density was sampled in vertical layers at every fifth point. As no systematic stratification of the snow density was found, SWE was finally calculated from snow depth and the average snow density computed from all measurements in a given year. At Bayelva, the snow density was generally confined to a range of $350\pm50\,\mathrm{kg\,m^{-3}}$ for all the surveys, while at Steinflåen plateau, the snow density was found to be around $450\,\mathrm{kg\,m^{-3}}$ in 2016. At Kvadehuksletta and Steinflåen plateau, the surveys were conducted along transects with regular sample intervals (see Figure 1), while a randomized array of sample points (see Gisnås et al., 2014, for details) was employed for Bayelva in most years, apart from the first years where we also had transects for Bayelva (see Table 1).





| Location | $\bar{z}$ [m a.s.l.] | $\sigma_z$ [m] | Area [km$^2$] | Survey years | Samples per survey |
|---|---|---|---|---|---|
| Bayelva | 23 | 9 | 0.5 | 2008, 2009, 2013-2016 | $853^t$, $617^t$, $105^r$ |
| Steinflåen plateau | 210 | 11 | 1.1 | 2016 | $45^t$ |
| Kvadehuksletta | 55 | 6 | 0.9 | 2016 | $30^t$ |

**Table 1.** Overview of study sites and snow surveys. $\bar{z}$ is the mean elevation and $\sigma_z$ is the standard deviation in the elevation both based on the Norwegian Polar Institute (2014) DEM. Key: t=transect, r=randomized array.

Basal ice layers occur in the area in conjunction with rain on snow events (Kohler and Aanes, 2004; Westermann et al., 2011a), and can constitute a major source of uncertainty for SWE measurements, both for single measurements and the resulting spatial distribution. In 2016, the depth of basal ice layers was measured using ice screws and their contribution to the SWE was accounted for. In addition, not accounted for internal ice layers and the spatial variability of average snow densities (see above) contribute to the uncertainty of the measurements. Finally, only a limited number of sampling points is considered, so that the obtained snow distributions are expected to deviate to a certain extent from the true snow distributions in the area (c.f. Gisnås et al., 2014).

In 2012, 2013 and 2016, an automatic time-lapse camera was deployed near the summit of Scheteligfjellet (694 m a.s.l.; c.f. Figure 1) overlooking the Bayelva site. The camera was a standard digital camera triggered by a Harbortronics time-lapse system, delivering daily images except for prolonged periods with low cloud cover. The raw camera images were orthorectified at a 1 m resolution and the snow cover was mapped for each pixel using a simple threshold on the grayscale intensity, so that fSCA could thus be determined for each image. The orthorectified images for most of the years are freely available through Westermann et al. (2015a).

In 2008, aerial images were obtained for the Bayelva site for five dates in June during the beginning of the snow melt period. For this purpose a digital camera was mounted to a UAV flying at an altitude of 100 to 250 m above ground which took between 700 and 1000 images per mission at nadir angles. As the images were taken in a near-random fashion over the entire area, fSCA was calculated by averaging over the fSCA determined for each image using a simple threshold criterion. Note that orthorectification of the images is not necessary since a sufficient number of images taken from random positions and/or altitudes is available. GPS-based surveys of remaining snow patches were available for five additional dates, so that a complete fSCA time series is available for the snow melt period in 2008.





## 3 Method

### 3.1 Simple snow model

To efficiently run a large number of ensemble members, a simple snow model (SSM) is employed, which computes snow melt rates according to surface energy balance formulations (as in the CryoGrid 3 ground thermal model; Westermann et al., 2016a).

The model is a blend of a single layer mass balance model, based on the UEB model (detailed in Tarboton and Luce, 1996; You et al., 2014) and a snow depletion curve parametrization based on the subgrid snow distribution submodel described in Liston (2004). Snow internal processes are not considered. In the following sections, we describe the governing equations of the SSM (see Table 2 for an explanation of constants).

### 3.1.1 Snow depletion curve

We use the probabilistic snow depletion curve formulations discussed in Liston (1999), Luce and Tarboton (2004) and Liston (2004) which parametrize the relationship between fSCA and SWE by using a probability density function (pdf) to represent the subgrid SWE distribution (SSD). A central assumption in these parametrizations is that the melt rate is uniform throughout the given grid cell. The relationship between the accumulated melt depth ($D_m$), the peak SSD pdf which we denote $f_P$, and the fSCA within the grid cell at time $t$ is given by

$$\text{fSCA}(t) = \int_{D_m(t)}^{\infty} f_P(D)\,dD\,. \tag{1}$$

Similarly the mean SWE depth is given by

$$\overline{D}(t) = \int_{D_m(t)}^{\infty} (D - D_m(t))\,f_P(D)\,dD\,. \tag{2}$$

Following Liston (2004) we parametrize the peak SSD using a two parameter lognormal pdf $f_P = f_P(D|\mu,\chi)$ where $\mu$ is the peak mean SWE and $\chi = \sigma/\mu$ is the peak subgrid coefficient of variation ($\sigma$ is the standard deviation). Our choice of parametric

distribution was motivated by independent measurements of the SSD which fit reasonably well to a lognormal distribution for most of the years. Equations (1) and (2) can both be solved analytically as presented in Liston (2004).

### 3.1.2 Mass and Energy Balance

To obtain the instantaneous net accumulation rate, $\mathcal{A}(t)$, we follow the UEB model through (You et al., 2014)

$$\mathcal{A}(t) = P(t) - M(t)\,, \tag{3}$$

where $P(t)$ is the precipitation rate and $M(t)$ is the melt rate. Sublimation is not considered as it is a relatively small contribution to the mass balance at our study area (Westermann et al., 2009). We use a linear transition to delineate between snowfall and rainfall as in You et al. (2014) with thresholds given in Table 2. We only consider rainfall as a positive contribution to





the mass balance during non-melting conditions when the rainwater generally refreezes in the snowpack (Westermann et al., 2011a). For melting conditions (where $D_m > 0$), we assume that rainfall becomes runoff directly.

The melt rate, $M$, is calculated based on a simplified snow energy balance. Here SSM differs from UEB in that we assume
that the snowpack is always isothermal and at $0°C$ under melting conditions. In this case, the energy balance becomes

$$Q_M(t) = Q_R^*(t) + Q_P(t) - Q_H(t) - Q_E(t) - Q_G(t), \tag{4}$$

where $Q_M$ is the snow melt flux, $Q_R^*$ is the global radiation, $Q_P$ is the heat advected by precipitation, $Q_H$ is the sensible heat flux, $Q_E$ is the latent heat flux and $Q_G$ is the ground heat flux. The last three fluxes are defined as positive when directed away from the surface and vice versa for the first two on the right hand side of (4). The global radiation is given by

$$Q_R^* = (1 - \alpha_S) S^\downarrow + L^\downarrow - \varepsilon_S \sigma_{SB} T_0^4, \tag{5}$$

in which $S^\downarrow$ and $L^\downarrow$ are the downwelling shortwave and longwave irradiances and the last term is the upwelling longwave radiation for the isothermal snowpack at $T_0 = 273.15\,K$. The snow albedo ($\alpha_S$) is parametrized prognostically through the continuous reset formulation (Dutra et al., 2010), which differentiates between accumulating, steady and ablating conditions as follows for a time increment $\Delta t$

$$\alpha_S(t+\Delta t) = \begin{cases} \alpha_S(t) + \min\left(1, \mathcal{A}(t)\Delta t/\tau_S\right)(\alpha_{\max} - \alpha_S(t)), & \mathcal{A}(t) > 0, \\ \max\left(\alpha_S(t) - \tau_A \Delta t, \alpha_{\min}\right), & \mathcal{A}(t) = 0, \\ (\alpha_S(t) - \alpha_{\min})\exp(-\tau_F \Delta t) + \alpha_{\min}, & \mathcal{A}(t) < 0, \end{cases} \tag{6}$$

where $\alpha_{\min}$ and $\alpha_{\max}$ are the minimum and maximum snow albedo values respectively, while $\tau_A$ and $\tau_F$ are aging (decay) rates for non-melting and melting snow respectively, and finally $\tau_S$ is a threshold daily snowfall which if exceeded leads to a reset of the snow albedo to its maximum value. This simple decay and reset type of snow albedo parametrization has been shown to perform reasonably well at our study site in the study of Pedersen and Winther (2005). The heat advected by rainfall, $Q_P$,
is diagnosed as in Tarboton and Luce (1996), while the turbulent fluxes of sensible ($Q_H$) and latent ($Q_E$) heat are evaluated through Monin-Obukhov similarity theory as presented in Westermann et al. (2016a). The ground heat flux is parametrized through a simple e-folding relationship during the melting period, i.e.

$$Q_G = Q_0 \exp\left(d_H t_m / z_E^2\right), \tag{7}$$

where $Q_0$ is the initial ground heat flux, $d_H$ is the thermal diffusivity of the ground and $z_E$ is the effective depth of the heat
transfer below the base of the snowpack, while $t_m$ is the number of days with melting conditions after peak accumulation. The initial ground heat flux is a free parameter determined in the assimilation, while $d_H$ is fixed based on the study of Westermann et al. (2009) and $z_E$ is set so that the ground heat flux decays to near zero a month into the melt season.



With all terms in the energy balance diagnosed the snow melt flux $Q_M$ can now be evaluated through eq. (4). We recall that an isothermal snowpack at $0°$C is assumed for (3) and (4), which is only justified for a melting snowpack: in this case, positive $Q_M$ correspond to melting and SWE reduction, while negative values correspond to refreezing of melt water and thus SWE increase. For a dry snowpack (as is generally the case before the snow melt), negative $Q_M$ values would lead to a cooling of the snowpack, which is not considered in this simple snow melt scheme. For simplicity, we work with a daily time step. To discard unphysical values (negative melt rates) we only consider days with net melting conditions, i.e. positive daily average snow melt fluxes. Thus, the daily averaged melt rate $M_n$ at day $n$ (lasting from $t_n$ to $t_{n+1}$) is obtained through

$$M_n = \max \left( \frac{1}{\rho_w L_f \Delta t} \int_{t_n}^{t_{n+1}} Q_M(t)\, dt, 0 \right), \tag{8}$$

where $\rho_w$ is the density of fresh water, $L_f$ is the latent heat of fusion and $\Delta t$ is the daily time step. It is worth emphasizing that with the formulation in (8) the effects of refreezing are still considered at subdaily time resolution. Similarly, the daily averaged precipitation rate is obtained through

$$P_n = \frac{1}{\Delta t} \int_{t_n}^{t_{n+1}} P(t)\, dt, \tag{9}$$

where we reiterate that $P(t)$ only includes the snowfall during the melt season. Now the daily averaged net accumulation rate is obtained through

$$\mathcal{A}_n = P_n - M_n, \tag{10}$$

and the accumulated melt depth $D_m$ is accounted for through

$$D_{m,n+1} = \max \left( D_{m,n} - \mathcal{A}_n \Delta t, 0 \right) H(\mu). \tag{11}$$

The peak mean SWE $\mu$ is updated via

$$\Lambda = \mu_n + \max \left( \mathcal{A}_n \Delta t - D_{m,n+1}, 0 \right), \tag{12}$$

through

$$\mu_{n+1} = \Lambda H(\Lambda - \tau_S), \tag{13}$$

where in both (11) and (13) the alternative Heaviside function is defined through $H(x) = 0$ if $x \leq 0$, and $H(x) = 1$ otherwise. Consequently, in (13) the peak mean SWE $\mu$ is only non-zero if $\Lambda$ exceeds the threshold $\tau_S$. Note that the formulation in (11) gradually resets the melt depth towards zero in the case of snowfall after the onset of melt following Liston (2004). This means that fSCA is not reset to unity in the case of new snowfall after a melting period unless the new snowfall leads to an increase in the peak SWE. In the study area, snowfall events occurred only rarely during the snow melt period and the new snow cover lasted only a short time. At sites where such events are more frequent, Durand et al. (2008) presents an alternative solution,



albeit at an increased computational expense. The initial conditions for the SSM are quite simple with the model integration being carried out independently for each hydrological year starting in the beginning of September where we assume that the surface is free of snow so that both $\mu$ and $D_m$ are initialized as zero. Moreover, both $\mu$ and $D_m$ are reset to zero following the complete disappearance of the snowpack within a grid cell defined (due to the infinite tail of $f_P$) as when fSCA $< 0.01$.

### 3.1.3 Forcing

Forcing terms in the form of precipitation, air temperature, relative humidity and wind speed, as well as downwelling longwave and shortwave radiation are required to diagnose the mass and energy balance in (3) and (4). These terms were obtained by downscaling ERA-Interim reanalysis data (Dee et al., 2011) at $0.75°$ resolution following Østby et al. (2017). This method

10 uses the linear theory of orographic precipitation in Smith and Barstad (2004) to downscale precipitation and a modification of the TopoSCALE approach described by Fiddes and Gruber (2014) for the remaining fields. The resulting values at $1\,\mathrm{km}$ and 6 hourly temporal resolution are further linearly interpolated in time to enable a stable computation of the time evolution of turbulent energy fluxes (using Monin-Obukhov similarity theory) following Westermann et al. (2016a). From these fluxes and the remaining surface energy balance fields, diurnally averaged melt rates are diagnosed. Similarly, averaged snowfall and

15 rainfall rates are obtained by taking the accumulated diurnal precipitation, normalizing and then delineating between rain and snow following You et al. (2014) as mentioned in the previous section.

| Symbol | Name | Value | Units (SI) | Reference |
|---|---|---|---|---|
| $\alpha_{\mathrm{max}}$ | Maximum snow albedo | 0.85 | $-$ | Dutra et al. (2010) |
| $\tau_S$ | Threshold snowfall | 0.01 | m (w.e.) | Dutra et al. (2010) |
| $\tau_F$ | Aging constant for melting snow | $2.78 \times 10^{-8}$ | $\mathrm{s}^{-1}$ | Dutra et al. (2010) |
| $\tau_A$ | Aging constant for non-melting snow | $9.26 \times 10^{-8}$ | $\mathrm{s}^{-1}$ | Dutra et al. (2010) |
| $T_R$ | Threshold temperature for rain | 276.15 | K | You et al. (2014) |
| $T_S$ | Threshold temperature for snow | 272.15 | K | You et al. (2014) |
| $\varepsilon_S$ | Emissivity of snow | 0.99 | $-$ | Westermann et al. (2016a) |
| $d_H$ | Thermal diffusivity of the ground | $6 \times 10^{-7}$ | $\mathrm{m}^2\,\mathrm{s}^{-1}$ | Westermann et al. (2009) |
| $z_E$ | Effective transfer depth | 1 | m | - |
| $\Delta t$ | Daily time step | 86400 | s | - |
| $L_f$ | Specific latent heat of fusion | $3.35 \times 10^5$ | $\mathrm{J\,kg}^{-1}$ | Tarboton and Luce (1996) |
| $\rho_w$ | Density of fresh liquid water | $10^3$ | $\mathrm{kg\,m}^{-3}$ | Tarboton and Luce (1996) |
| $\sigma_{\mathrm{SB}}$ | Stefan-Boltzmann constant | $5.67 \times 10^{-8}$ | $\mathrm{W\,m}^{-2}\,\mathrm{K}^{-4}$ | Tarboton and Luce (1996) |

**Table 2.** List of constants used in the simple snow model runs along with the corresponding reference studies.



## 3.2 Satellite retrievals

We make use of satellite retrievals between May and September which contained the snow melt period in all the investigated years. Only observations that fell inside the melt season were assimilated as this is where information about the true snow cover depletion is contained. Due to frequent cloud cover, the effective revisit frequency of fSCA retrievals is irregular, with
prolonged data gaps occurring regularly.

### 3.2.1 MODIS

We employ version 006 of the level 3 daily 500 meter resolution fSCA retrievals from the Moderate Resolution Imaging Spectroradiometer (MODIS) on board the satellites Terra (MOD10A1 product; Hall and Riggs, 2016a) and Aqua (MYD10A1 product; Hall and Riggs, 2016b). The retrieval algorithm is based on an linear fit of the normalized difference snow index
(NDSI) measured by MODIS to fSCA retrievals from 'ground truth' LandSat scenes as described in Salomonson and Appel (2004, 2006). The NDSI exploits the fact that snow is highly reflective in the visible, but a good absorber in the shortwave infrared which sets it apart from other natural surfaces such as clouds, vegetation and soil (Painter et al., 2009).

We average over all the pixels considered for each study area as shown in Figure 1. If both Terra and Aqua MODIS fSCA
retrievals are available for a given day, the Terra MODIS retrievals are prioritized due to faulty Aqua MODIS band 6 detectors and the subsequent use of band 7 in the NDSI causing greater pixel misregistration (Salomonson and Appel, 2006). If only Aqua MODIS fSCA retrievals are available for a given pixel, these are still used to maximize the temporal coverage of the snow cover depletion. Despite small deviations in the measurement footprint (Figure 1), we compare MODIS fSCA retrievals to the ground truth obtained from the automatic camera system, UAV surveys and ground based measurements (Section 2.2).
From this comparison, we estimate a RMSE of $\sigma_{\mathrm{MOD}} = 0.13$ for the MODIS retrievals, which is used for the observation error covariance matrix in the assimilation step (Section 3.3.2).

### 3.2.2 Sentinel-2

For the year 2016, we complement the MODIS fSCA retrievals with aggregated $20\,\mathrm{m}$ resolution retrievals from the Sentinel-2A and 2B missions (Drusch et al., 2012). fSCA estimates are derived from the level 1C orthorectified top of the atmosphere
(TOA) reflectance product, with cloud-free scenes manually selected. For this purpose, NDSI is computed from reflectances ($r$) from a visible (b3, centered on $0.56\,\mu\mathrm{m}$) and a shortwave infrared band (b11, centered on $1.61\,\mu\mathrm{m}$) through

$$\mathrm{NDSI_{S2}} = \frac{r_{\mathrm{b3}} - r_{\mathrm{b11}}}{r_{\mathrm{b3}} + r_{\mathrm{b11}}} . \tag{14}$$

As b3 is at $10\,\mathrm{m}$ resolution, pixels are aggregated to the $20\,\mathrm{m}$ resolution of b11. Each pixel is then either classified as snow covered (NDSI$\geq 0.4$) or snow free (NDSI$< 0.4$) where the NDSI threshold was chosen in line with the work of Hall et al.
(2002). The binary (snow/no snow) pixels are then aggregated to the approximate footprint of the independent snow surveys conducted at each site (Figure 1) to obtain Sentinel-2 derived fSCA estimates. The retrieval process is illustrated schematically





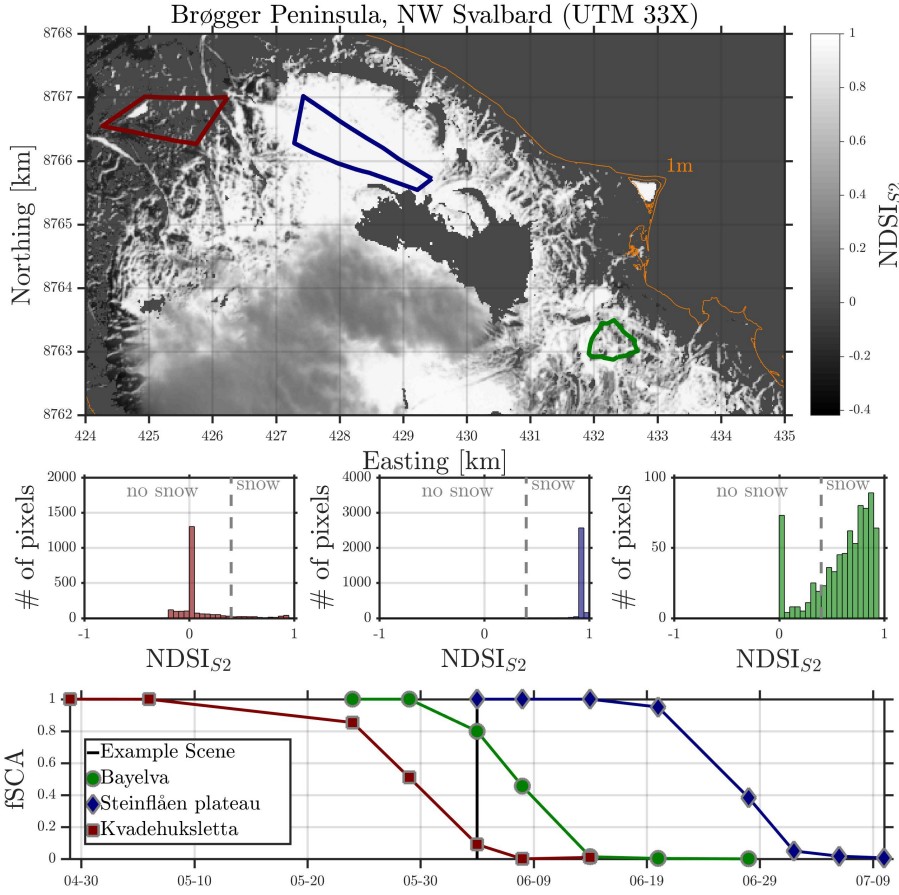

**Figure 2.** Top panel: Sentinel-2 NDSI estimates from an example scene (taken 04.06.2016) over the Brøgger Peninsula with Kvadehuksletta, Steinflåen plateau and Bayelva marked with red, green and blue polygons; coastline in orange. Middle panels: NDSI histograms of the same example scene (Kvadehuksletta: left, Steinflåen plateau: middle; Bayelva: right) with the threshold at NDSI= 0.4 marked. Bottom panel: Time series of Sentinel-2 NDSI-based fSCA retrievals for the 2016 melt season.

in Figure 2. By comparing the Sentinel-2 retrievals to the ground truth fSCA from the automatic camera system in 2016 we estimate a RMSE of $\sigma_{S2} = 0.09$, which is used for the observation error covariance matrix in the assimilation step.

### 3.3 Ensemble Data Assimilation

In the following section we will first describe how the prior ensemble of model realizations is setup and subsequently how
5    this is updated to a posterior ensemble through the assimilation of Terra/Aqua MODIS and Sentinel-2 fSCA retrievals using ensemble-based batch smoother schemes. Our approach is based on the probabilistic SWE reconstruction techniques outlined in the work of Durand et al. (2008), Girotto et al. (2014b) and Margulis et al. (2015) with all the assimilation schemes being traced back to the work of Van Leeuwen and Evensen (1996). The novelty of this work is the direct recovery of the peak subgrid





SWE distribution using a novel land surface data assimilation scheme adopted from the petroleum reservoir history matching community (Emerick and Reynolds, 2013) in conjunction with analytical Gaussian anamorphosis (Bertino et al., 2003).

### 3.3.1 Ensemble Generation

The prior ensemble of SSM realizations is generated by independently drawing parameter values from the distributions listed in Table 3. These parameters are held constant throughout the integration of the SSM. Two of these are multiplicative bias parameters that perturb the mass balance through the net accumulation rate

$$\mathcal{A}_{n,j} = b_{P,j} P_n - b_{M,j} M_{n,j},$$

for $j \in 1 : N_e$ where $N_e$ is the number of ensemble members. We inherently assume the model forcing to be the major source of uncertainty (De Lannoy et al., 2010; Raleigh et al., 2015), and that the error in the forcing can be modeled through constant multiplicative biases in the mass balance. Consequently, each of these bias parameters is modeled as a positive definite lognormal random variable. This is in line with the perturbations in Girotto et al. (2014b) on the precipitation rate, but we also perturb the melt rate. Moreover, we assume that the ensemble of net accumulation rates is on average unbiased due to the applied downscaling method (Østby et al., 2017) and thus assign the two bias parameters a mean of unity. The precipitation rates are also perturbed by the same bias parameter in the computation of the heat advected by precipitation ($Q_P$) in the surface energy balance that contributes to the melt rate $M_n$.

| Symbol | Name | Distribution | Support | Mean | Variance | Units |
|--------|------|--------------|---------|------|----------|-------|
| $\chi$ | Coefficient of Variation | Logit-normal | $(0, 0.8)$ | 0.4 | 0.01 | - |
| $Q_0$ | Initial ground heat flux | Logit-normal | $(0, 40)$ | 20 | 20 | $\mathrm{Wm}^{-2}$ |
| $\alpha_{\min}$ | Minimum snow albedo | Logit-normal | $(0.45, 0.55)$ | 0.5 | 0.02 | - |
| $b_P$ | Precipitation bias | Lognormal | $(0, \infty)$ | 1 | 0.04 | - |
| $b_M$ | Melt bias | Lognormal | $(0, \infty)$ | 1 | 0.01 | - |

**Table 3.** Overview of distributions from which the prior ensemble of parameters are independently drawn.

In addition to the mass balance forcing, the peak subgrid coefficient of variation $\chi$ (Section 3.3.1) is a source of uncertainty. Here our prior assumption is that $\chi$ is on average close to the value given in the dichotomous key (simple classification scheme) of Liston (2004). For the tundra environment of our study sites this amounts to a mean value of $0.4$. In addition, we assume that $\chi$ is double bounded between 0 and 0.8, with negative values being unphysical and the upper bound close to the maximum value in the key of Liston (2004), and thus model it as a logit-normal random variable. Likewise, both the initial ground heat flux at the onset of melt $Q_0$, and the minimum snow albedo $\alpha_{\min}$ are uncertain and we model these parameters as double bounded logit-normal random variables (see Table 3 for values).




The logit transform for a variable $x$ bounded between $a$ and $b$ is given by

$$\widetilde{x} = \mathrm{logit}_{(a,b)}(x) = \ln\left(\frac{x-a}{b-a}\right) - \ln\left(1 - \frac{x-a}{b-a}\right),\tag{15}$$

while the inverse transform is given by

$$x = \mathrm{logit}_{(a,b)}^{-1}(\widetilde{x}) = a + (b-a)/\left(1 + e^{-\widetilde{x}}\right).\tag{16}$$

To generate a prior ensemble of a logit-normally distributed random variable: we apply the logit transform to the mean, add $N_e$ ensemble members of Gaussian white noise with a consistent variance and then apply the inverse transform. We emphasize that by perturbing the forcing, ground heat flux and coefficient of variation and performing an ensemble integration of SSM we also get an ensemble of states (fSCA$_{n,j}$, $D_{m,n,j}$, $\bar{D}_{n,j}$), remaining parameters ($\mu_j$) and pdfs ($f_{P,j}$) that are consistent with the fixed prior parameter ensemble.

### 3.3.2    Batch Smoothers

Herein we will describe the main batch smoother scheme used in the assimilation, namely the ES-MDA, in addition to the ES and the PBS that are used for comparison. In a batch smoother all the observations, in this case all fSCA retrievals from the snow cover depletion during a given melt season, are assimilated at once in a single batch (Dunne and Entekhabi, 2005), as opposed to sequentially as for a filter (Bertino et al., 2003). For clarity, we follow as closely as possible the conventional

notation in the data assimilation literature as laid out in Ide et al. (1997). Consequently, we let $N_e$, $N_o$, $N_a$, $N_s$, $N_p$ and $N_t$ denote the number of ensemble members, observations, assimilation cycles, states, parameters and time steps during an annual (September-August) model integration. Subsequently, we let $\mathbf{X}$ denote the $(N_s \times N_t) \times N_e$ matrix containing the ensemble of states and $\mathbf{\Theta}$ is the $N_p \times N_e$ matrix containing the ensemble of parameters. Moreover, we let $\mathbf{y}$ denote the $N_o \times 1$ vector of observations containing all available fSCA observations during the ablation season, $\mathbf{Y}$ is the $N_o \times N_e$ matrix containing

the ensemble of perturbed fSCA observations, $\mathbf{H}$ is the observation operator (mapping from the state space to the observation space) and $\mathbf{R}$ is the $N_o \times N_o$ observation error covariance matrix.

The **ensemble smoother with multiple data assimilation** (ES-MDA Emerick and Reynolds, 2013) is an iterative scheme, requiring multiple ensemble model integrations and analysis steps. The process of collecting the perturbed and predicted

observations during the ensemble integration into a batch and performing the analysis step is referred to as one assimilation cycle and we will let the current iteration number be denoted as $\ell$. In such a case, the ES-MDA scheme is set up as follows, for $\ell \in 0 : N_a$ iterations:

   1. Run an ensemble model integration, that is for $n \in 0 : (N_t - 1)$ time steps compute

$$\mathbf{X}_{n+1}^{(\ell)} = \mathcal{M}\left(\mathbf{X}_n^{(\ell)}, \mathbf{\Theta}^{(\ell)}\right),\tag{17}$$

where $\mathcal{M}$ is the SSM operator defined through equations (1), (2), (11) and (13).



2. Collect the batch of predicted observations

$$\widehat{\mathbf{Y}}^{(\ell)} = \mathbf{H}\mathbf{X}^{(\ell)}, \tag{18}$$

and perturbed observations

$$\mathbf{Y}^{(\ell)} = \mathbf{y} \otimes \mathbf{1}^T + \sqrt{\alpha^{(\ell)}}\mathbf{R}^{1/2}\boldsymbol{\epsilon}^{(\ell)}, \tag{19}$$

where $\otimes$ is the outer product, $\mathbf{1}$ is an $N_o \times 1$ vector of ones, the $^T$ superscript denotes the transpose, $\alpha^{(\ell)}$ is the observation error inflation coefficient and $\boldsymbol{\epsilon}^{(\ell)}$ is a $N_o \times N_e$ matrix containing zero mean Gaussian white noise with unit variance.

3. If $\ell < N_a$, *otherwise stop the algorithm*, transform the parameters using appropriate analytical Gaussian anamorphosis functions $\psi$ (Bertino et al., 2003)

$$\widetilde{\boldsymbol{\Theta}}^{(\ell)} = \psi\left(\boldsymbol{\Theta}^{(\ell)}\right), \tag{20}$$

where $\psi$ is the natural logarithm transform for the bias parameters and the logit transform (15) for the remaining parameters.

4. Perform the Kalman-like analysis step in the transformed space

$$\widetilde{\boldsymbol{\Theta}}^{(\ell+1)} = \widetilde{\boldsymbol{\Theta}}^{(\ell)} + \mathcal{C}_{\widetilde{\boldsymbol{\Theta}}\widehat{\mathbf{Y}}}^{(\ell)}\left(\mathcal{C}_{\widehat{\mathbf{Y}}\widehat{\mathbf{Y}}}^{(\ell)} + \alpha^{(\ell)}\mathbf{R}\right)^{-1}\left(\mathbf{Y}^{(\ell)} - \widehat{\mathbf{Y}}^{(\ell)}\right), \tag{21}$$

where the transformed parameter-predicted observation error covariance matrix and the predicted observation error covariance matrices are given by

$$\mathcal{C}_{\widetilde{\boldsymbol{\Theta}}\widehat{\mathbf{Y}}}^{(\ell)} = \frac{1}{N_e}\widetilde{\boldsymbol{\Theta}}^{(\ell)'}\widehat{\mathbf{Y}}^{(\ell)'T}, \tag{22}$$

and

$$\mathcal{C}_{\widehat{\mathbf{Y}}\widehat{\mathbf{Y}}}^{(\ell)} = \frac{1}{N_e}\widehat{\mathbf{Y}}^{(\ell)'}\widehat{\mathbf{Y}}^{(\ell)'T}, \tag{23}$$

respectively, in which primes ($'$) denote anomalies (deviations from the ensemble mean).

5. Apply the appropriate inverse transforms to recover the updated parameters

$$\boldsymbol{\Theta}^{(\ell+1)} = \psi^{-1}\left(\widetilde{\boldsymbol{\Theta}}^{(\ell+1)}\right), \tag{24}$$

where $\psi^{-1}$ is the exponential transform for the bias parameters and the inverse logit transform (16) for the remaining parameters.

The **ensemble smoother** (ES; Van Leeuwen and Evensen, 1996) scheme is recovered in the algorithm above for $N_a = \alpha^{(\ell)} = 1$, that is without inflating the observation error covariance and in the absence of iterative analysis steps. This is the



algorithm that was used in the probabilistic SWE reconstruction scheme of Durand et al. (2008) and Girotto et al. (2014b). It is the observation error inflation coefficient $\alpha^{(\ell)}$ in (21), along with the iterations, that sets the ES-MDA scheme apart from the traditional ES scheme. The idea with the ES-MDA is to perform multiple smaller analysis steps as opposed to one abrupt analysis step. In the case of a non-linear models this is expected to yield a better approximation of the true posterior (Emerick and Reynolds, 2013). For the ES-MDA to give a nearly unbiased estimate (c.f. Stordal and Elsheikh, 2015) it is required that the coefficients satisfy $\sum_{\ell=0}^{N_a-1} \frac{1}{\alpha^{(\ell)}} = 1$. In our case this is accomplished by setting all the coefficients as $\alpha^{(\ell)} = N_a$ and specifying $N_a$ before any assimilation cycles are carried out. As an alternative that is not pursued here an adaptive form of the scheme is presented in Le et al. (2016) where the inflation coefficients are calculated on the fly based on a cost function and the iterations stop once the algorithm converges. It is worth emphasizing that the analysis step (21) only updates the parameters and a consistent ensemble of states are found from the subsequent ensemble model integration. As mentioned, the parameter matrix $\widetilde{\Theta}$ in (21) is transformed through analytical Gaussian anamorphosis (Bertino et al., 2003) to ensure that the priors are Gaussian in which case the ensemble smoother-based analysis step is variance minimizing in the case of a linear model (Van Leeuwen and Evensen, 1996). The entire methodology, with ES-MDA as the DA scheme, is depicted in Figure 3.

In the **particle batch smoother** (PBS), as introduced for snow assimilation by Margulis et al. (2015), an initial ensemble integration of the model is carried out to obtain the prior state and parameter estimates. A priori, each particle (i.e. ensemble member; Van Leeuwen, 2009) is given an equal weight of $1/N_e$. Subsequently, following Margulis et al. (2015), the normalized posterior importance weights $w_j \in [0,1]$ are diagnosed through the expression

$$w_j = p\left(\mathbf{y}|\widehat{\mathbf{X}}_j\right) p\left(\widehat{\mathbf{X}}_j\right) / \sum_{j=1}^{N_e} \left(p\left(\mathbf{y}|\widehat{\mathbf{X}}_j\right) p\left(\widehat{\mathbf{X}}_j\right)\right), \tag{25}$$

where $\widehat{\mathbf{X}}_j = [\mathbf{X}_j \, ; \, \Theta_j]$ is the augmented state vector for the $j$-th particle and the Gaussian likelihoods are given by

$$p\left(\mathbf{y}|\widehat{\mathbf{X}}_j\right) = c_0 \exp\left[-0.5\left(\mathbf{y} - \widehat{\mathbf{Y}}_j\right)^{\mathrm{T}} \mathbf{R}^{-1}\left(\mathbf{y} - \widehat{\mathbf{Y}}_j\right)\right]. \tag{26}$$

Note in particular that (25) is a direct application of Bayes' rule with the normalizing denominator having two important consequences. Firstly, $c_0 = 1/\sqrt{(2\pi)^{N_o}|\mathbf{R}|}$ cancels out thus avoiding errors introduced through floating point arithmetic ($(2\pi)^{N_o}$ is a relatively large number). Secondly, the prior weights $p(\widehat{\mathbf{X}}_j)$ also fall out as they are equal ($1/N_e$) for all particles. Thereby, with this choice of likelihood, the analysis step (25) simplifies to a simple ratio of likelihoods

$$w_j = \frac{\exp\left[-0.5\left(\mathbf{y} - \widehat{\mathbf{Y}}_j\right)^{\mathrm{T}} \mathbf{R}^{-1}\left(\mathbf{y} - \widehat{\mathbf{Y}}_j\right)\right]}{\sum_{j=1}^{N_e} \exp\left[-0.5\left(\mathbf{y} - \widehat{\mathbf{Y}}_j\right)^{\mathrm{T}} \mathbf{R}^{-1}\left(\mathbf{y} - \widehat{\mathbf{Y}}_j\right)\right]}, \tag{27}$$

where the posterior weights $w_j$ sum to unity. The posterior ensemble will still span the range of the prior ensemble, as the analysis step only changes the relative weights of the ensemble members and not their position within the state and parameter space. Through the ranking of these weights the marginal cumulative distribution functions for all the state variables and parameters




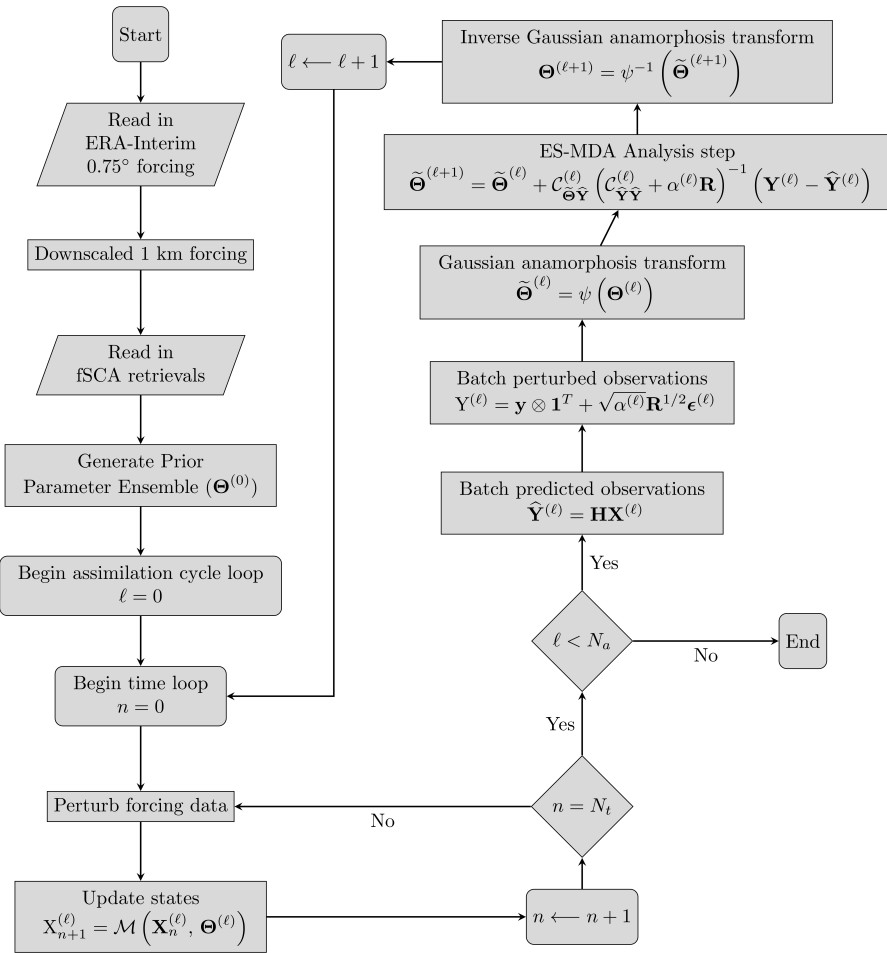

**Figure 3.** Flowchart depicting the methodology with the ES-MDA as the DA scheme. Symbols defined in the text.

are readily recovered thus enabling a diagnosis of quantile values. Note that the PBS is equivalent to running a particle filter without re-sampling and using the prior as the importance density (see Van Leeuwen, 2009). Consequently, the PBS is in itself not novel as a data assimilation scheme; in fact it is very similar to the generalized likelihood uncertainty estimation method (GLUE; Beven and Binley, 1992) with the choice of a formal Gaussian likelihood function. As discussed in Van Leeuwen
5    (2009), due to the absence of re-sampling, even for medium dimensional systems with a large number of observations to be assimilated, we expect the PBS to be degenerate with only a few particles carrying the majority of the importance weights. Nevertheless, a major advantage of the PBS is its computational efficiency, requiring only one ensemble model integration and one efficient analysis step (27). In this study, the PBS and the ES are used to benchmark the performance of the ES-MDA.



## 4 Results

### 4.1 Interannual variability and comparison to in-situ measurements

Here, we present results of the ES-MDA scheme with 100 ensemble members and 4 assimilation cycles (Section 3.3.2) for all
the years and sites where snow surveys were conducted. Figure 4 shows the time series of the prior and posterior fSCA (left
panel) and mean SWE, $\overline{D}$, (right panel) estimates, as well as satellite (left panel) and snow survey observations (right panel). In
most years, the assimilation brings the posterior estimate closer to the observed fSCA and considerably constrains the spread
of the ensemble compared to the prior. For some occasions, such as Bayelva 2008, Bayelva 2009 and Kvadehuksletta 2016, the
timing of the snow melt in the prior is significantly biased by as much as three weeks compared to the observations. Even if the
prior ensemble does not encompass the observations, the iterative ES-MDA scheme allows the posterior to converge towards
the fSCA observations (left panels), leading to much improved SWE estimates (right panels). In other occasions, such as 2015
at Bayelva and 2016 at Steinflåen plateau, the prior ensemble is a reasonable estimate and the assimilation merely constrains
the spread of the ensemble and adjusts the median slightly. Both for Bayelva in 2015 and Kvadehuksletta in 2016, some of the
initial fSCA observations, which indicate a slight ablation, are completely ignored by the assimilation, as this early onset of
melt is inconsistent with the model - even when biases are accounted for. This ablation could be real and due to processes not
accounted for in the model, such as wind erosion.

In-situ measurements of peak mean SWE are available for years with low (2008, 2016), medium (2013, 2015) and high
(2009, 2014) values of SWE, ranging from 0.08 m w.e. (Kvadehuksletta 2016) to 0.48 m w.e. (Bayelva 2014). With the ex-
ception of two cases (Bayelva 2013 and Steinflåen plateau 2016), the assimilation brings the ensemble median closer to the
observed peak mean SWE, while at the same time constraining the spread of the ensemble. We emphasize that the assimilation
performs a global bias correction for peak SWE. This is especially evident for Kvadehuksletta 2016 where the assimilation
unrealistically truncates the duration of the snow season as a result of a strong correction in the positive bias. Both in 2008
and 2009 for Bayelva, ES-MDA shifts the estimates to better match in-situ SWE observations (which were not assimilated),
despite the prior range being far from the observations. In general, the posterior ensemble peak mean SWE is remarkably close
to the observed SWE although the timing is not perfect as on many occasions the model predicts some accumulation after the
snow surveys were conducted.

Figure 5 displays the prior, posterior and observed subgrid SWE distributions (SSDs) for the years and sites in which in-situ
snow surveys are available. Again, with the exception of Bayelva 2013 and Steinflåen plateau 2016, the assimilation brings the
mean of the peak SSD closer to the observations. The agreement between the posterior and observed mean value is striking
for quite a few years and sites such as 2009 and 2014 for Bayelva and 2016 for Kvadehuksletta. Furthermore, the shapes of
the observed and posterior distributions agree well, e.g. for Bayelva 2008, Bayelva 2013 and Bayelva 2016. Once more, the
correction from prior to posterior is largest for Bayelva 2008 and Bayelva 2009 for which the prior model fSCA was furthest
from the satellite retrievals. As for the prior ensemble SSD, with the exception of Bayelva 2013 where it is a good estimate,





**Figure 4.** Results for the ES-MDA with $N_a = 4$ and $N_e = 10^2$; prior (red) and posterior (blue) fSCA (first column) and average SWE $\overline{D}$ (second column) as a function of time (months on the $x$-axis); shading: $90^{th}$ percentile range of the ensemble; solid lines: ensemble median; yellow dots: assimilated MODIS and Sentinel-2 fSCA retrievals; black diamonds: non-assimilated average SWE from snow surveys.

it is generally too positively skewed (i.e. has a long tail) compared to the observed SSD. There are a few occasions where the match between the posterior and observed SSDs is poor, such as Steinflåen plateau 2016 and Bayelva 2015. We conclude that the analysis in most occasions significantly improves the fit between modeled and observed snow distributions. Some of the



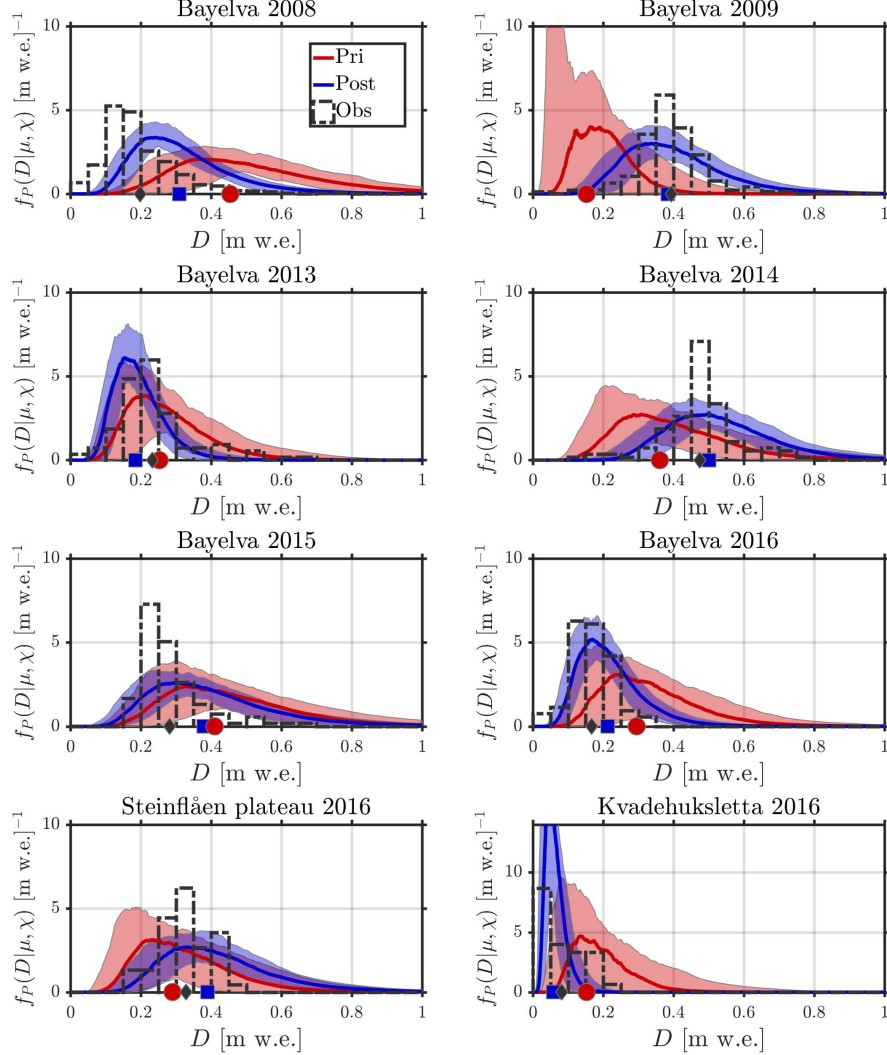

**Figure 5.** Prior (red), ES-MDA ($N_a = 4$, $N_e = 10^2$) posterior (blue) and observed (from snow surveys; dashed black) subgrid SWE distributions (probability density as a function of subgrid SWE); shaded areas: 90th percentile range of the ensemble; solid lines: ensemble median; markers: mean value.

observed distributions, such as that for Kvadehuksletta 2016, are hard to match as they do not conform well to a lognormal distribution, most likely due to the limited number of sample points (Section 2.2).



## 4.2 Evaluation of data assimilation schemes

| Symbol | Prior | | | ES-MDA | | | ES | | | PBS | | |
|---|---|---|---|---|---|---|---|---|---|---|---|---|
| | Bias | RMSE | $R^2$ | Bias | RMSE | $R^2$ | Bias | RMSE | $R^2$ | Bias | RMSE | $R^2$ |
| fSCA (106) | 0.21 | 0.02 | 0.80 | 0.03 | 0.01 | 0.97 | 0.03 | 0.01 | 0.97 | 0.03 | 0.01 | 0.97 |
| $\mu$ (8) [m w.e.] | 0.13 | 0.21 | 0.01 | 0.06 | 0.09 | 0.77 | 0.06 | 0.12 | 0.76 | 0.06 | 0.08 | 0.76 |
| $\chi$ (8) | 0.01 | 0.16 | 0.00 | $-0.02$ | 0.13 | 0.37 | 0.02 | 0.14 | 0.33 | $-0.03$ | 0.13 | 0.06 |

**Table 4.** Summary of evaluation metrics, i.e. bias, time averaged ensemble RMSE and square correlation coefficient ($R^2$), for the the independently observed variables/parameters fSCA, peak SWE ($\mu$) and peak subgrid coefficient of variation ($\chi$). Number of observations in brackets next to the corresponding symbols. All the metrics are averaged over 100 independent runs each with 100 ensemble members. The ES-MDA was run with $N_a = 4$ assimilation cycles.

In addition to the ES-MDA scheme, we evaluate the PBS and ES (Section 3.3.2) with regards to in-situ measurements, using an ensemble size of 100 members for all schemes. Three error metrics are considered: the bias (mean error), RMSE and the square correlation coefficient, the results are summarized in Table 4. For the fSCA, all the schemes achieved a major improve-
5    ment relative to the prior with an almost tenfold reduction in bias, a halving of RMSE and an almost perfect correlation to the in-situ fSCA retrievals. For the peak mean SWE, the PBS performs best in terms of RMSE and bias, followed closely by ES-MDA which, in turn, has the highest correlation coefficient. With regards to the peak subgrid coefficient of variation, ES-MDA performs best across all the metrics, tying with ES for (absolute) bias and the PBS for RMSE. As considerably more in-situ observations are available for fSCA than for SWE, the evaluation for fSCA must be considered more robust.

Observed, prior and posterior peak mean SWE and peak subgrid coefficient of variation for different years/sites are shown in Figure 6. As discussed in Section 4.1, the assimilation tends to move the posterior peak mean SWE estimates closer to the observed mean SWE when compared to the prior. However, clear performance differences are found between the different schemes for a number of situations: in 2008, the PBS is not able to correct for as much of the bias in the peak mean SWE
15    compared to the ES-MDA and the ES. For the remaining years the performance of the schemes in terms of estimating peak mean SWE is quite similar, but the spread of the ES is by far the largest overall, followed by the PBS and the ES-MDA. The PBS ensemble shows indications of degeneracy for some years (e.g. 2008 and 2009) where the median coincides with either the 5[th] or 95[th] percentile value. This indicates that the majority of the weight in the PBS is carried by just a few ensemble members (or particles). For the coefficient of variation, the 90[th] percentile range of the ES-MDA posterior ensemble encompassed
20    the observed value, with two exceptions (Steinflåen plateau 2016 and Kvadehuksletta 2016), while this is not (as much) the case for the ES (three exceptions) and the PBS (five exceptions). The performance differences explain the higher correlation





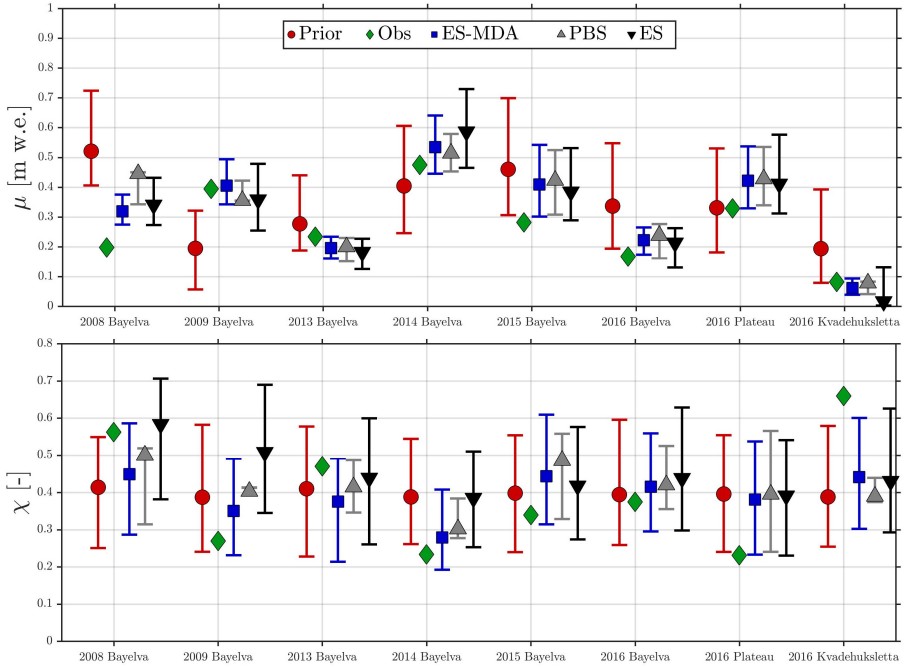

**Figure 6.** Prior median, observed, ES-MDA (with $N_a = 4$) posterior median, PBS posterior median and ES posterior peak mean SWE $\mu$ (upper panel) and peak subgrid coefficient of variation $\chi$ (lower panel); error bars: 90[th] percentile range; all DA schemes were run with $10^2$ ensemble members.

coefficient for the coefficient of variation for the ES-MDA scheme in Table 4. Similar to peak mean SWE, PBS shows signs of degeneracy (e.g. Bayelva 2009, complete degeneracy) for the coefficient of variation estimation. On some occasions (e.g. Bayelva 2008, Bayelva 2009 and Bayelva 2014), the posterior ensemble median is effectively pulled closer to the observed coefficient of variation when compared to the prior. Moreover, on the same occasions the ensemble spread is slightly constrained.

5   When compared to the peak mean SWE, however, it seems that it is much harder to constrain estimates of the coefficient of variation regardless of scheme, although it is possible to shift the ensemble in the right direction.

We gauged the sensitivity of the three batch smoother schemes with respect to ensemble size and the number of assimilation cycles by considering the fractional improvement (FI) in RMSE that was achieved through the analysis step for all available

10  'ground truth' observations. Figure 7 demonstrates the differences in performance with regards to this metric. On the one hand, the PBS requires an ensemble size of 1000 to converge to a stable FI of around 75%, 20% and 60% for the fSCA, peak subgrid coefficient of variation, and peak mean SWE respectively. On the other hand, ES-MDA with 4 assimilation cycles converges at already 100 ensemble members at a similar FI to the PBS. The ES performs worst regardless of ensemble size, with an FI of around $70\%$, $10\%$ and $55\%$ even with $10^5$ ensemble members, while requiring 100 ensemble members for convergent results.

15  For all schemes, the greatest improvements were achieved for fSCA, followed by peak mean SWE, while by far the lowest



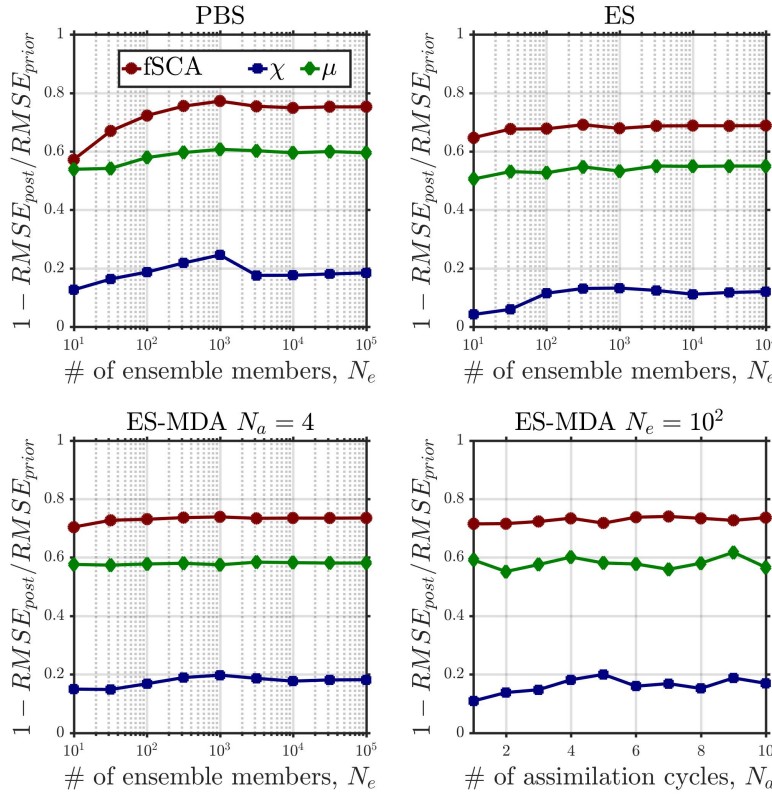

**Figure 7.** Fractional improvement in time averaged ensemble RMSE through the analysis step (1 being perfect and 0 no effect) as a function of the number of ensemble members for the fSCA, peak mean SWE $\mu$ and coefficient of variation $\chi$; top left: particle batch smoother, PBS; top right; ensemble smoother, ES; bottom left; ensemble smoother with multiple data assimilation, ES-MDA; bottom right: FI as a function of assimilation cycles in the ES-MDA. The FI for $N_e \leq 100$ are averaged over 100 independent ensemble model integrations.

improvements were achieved for the peak subgrid coefficient of variation. With an ensemble size of 100, the ES-MDA required around 4 assimilation cycles to converge to a stable performance, i.e. there is no marked improvement of FI for more cycles (Figure 7, bottom right). Therefore, the results for ES-MDA must be considered particularly favorable, as stable performance is achieved with only 4 assimilation cycles and 100 ensemble members.



## 4.3 Effects of observation error

| | | MODIS | | | | MODIS+S2 | | | |
|---|---|---|---|---|---|---|---|---|---|
| Symbol | # of obs | FI Bias | FI RMSE | $R^2_{prior}$ | $R^2_{post}$ | FI Bias | FI RMSE | $R^2_{prior}$ | $R^2_{post}$ |
| $\mu$ [m w.e.] | 3 | 0.61 | 0.62 | 0.67 | 1.00 | 0.61 | 0.62 | 0.69 | 1.00 |
| $\chi$ | 3 | -0.77 | 0.11 | 0.00 | 0.41 | -0.60 | 0.15 | 0.00 | 0.48 |

**Table 5.** Summary of evaluation metrics, i.e. fractional improvement in bias and fractional improvement in time averaged ensemble RMSE as well as prior and posterior square correlation coefficient ($R^2$), using the ES-MDA ($N_e = 10^2$, $N_a = 4$) assimilating only MODIS as well as assimilating MODIS and Sentinel-2 observations for all the independently observed variables/parameters in 2016. All the metrics are averaged over 100 independent runs each with 100 ensemble members.

The effects of observation error were studied by running the ES-MDA ($N_e = 10^2$, $N_a = 4$) and assimilating first only MODIS observations and then both MODIS and Sentinel-2 observations, the latter with significantly lower observation error (Section 3.2), for the 2016 snow season at all study sites. As previously discussed the Sentinel-2 fSCA retrievals are based on higher

resolution optical reflectance retrievals. As such they are expected to be less prone to representativeness error (i.e. error due to differences in the measurement footprint), and thus observation error, in that the area in which the snow surveys were conducted is more accurately covered by the retrievals. Furthermore, the Sentinel-2 scenes used for fSCA retrievals were manually checked to be cloud free, this was not the case for the MODIS scenes. Table 5 summarizes various performance metrics for the two different runs. For the peak mean SWE depth ($\mu$), there is no difference when including Sentinel-2 fSCA

retrievals in the assimilation. For the coefficient of variation, however, there is a noticeable increase in FI for both the bias and the RMSE, as well as a considerable increase in the correlation coefficient. This improvement in the coefficient of variation estimation is expected because upon including Sentinel-2 fSCA retrievals with lower observation error the shape of the snow depletion curve is more constrained. Since the shape of the depletion curve is closely tied to the value of the peak subgrid coefficient of variation ($\chi$), the inclusion of even just a few Sentinel-2 fSCA retrievals with lower error might add considerable

information to better constrain the value of $\chi$.

## 5 Discussion

### 5.1 Interannual variability and comparison to in-situ measurements

For all considered years and sites, the ES-MDA scheme both brings the ensemble median fSCA closer to the observed fSCA and significantly constrains the spread of the ensemble (Figure 4). Thus, the posterior effectively fills the gaps in the remotely

sensed fSCA time series using a physically based snow model which is bias-corrected through the assimilation, while at the




same time accounting for uncertainties in the retrievals. In addition, ES-MDA is generally able to correct the prior estimates of the peak mean SWE towards the independently observed values, which is essentially achieved through a bias correction on the forcing. Even if the downscaled forcing is biased it is likely a better model input than forcing data obtained directly from coarse-scale reanalyses (Østby et al., 2017). For example, the lapse rate correction on temperature in the downscaling (c.f.
Fiddes and Gruber, 2014) will adversely affect the precipitation and melt rates at the more elevated Steinflåen plateau. This effect is not captured in the reanalysis product (Dee et al., 2011) where the elevation of the nearest grid point is near sea level.

Figure 5 shows that for most of the years the prior median is a poor estimate of the observed peak mean SWE. This has important consequences for SWE modeling as it indicated that an open-loop (i.e. no DA) run would have a tendency to not
be representative at the grid scale. One reason for this discrepancy is that many crucial processes such as wind drift are not included in our model. Moreover, the subgrid variability of the SWE was typically overestimated in the prior, with the prior distributions typically being too skew. It would be possible to run a more sophisticated model (e.g. ALPINE3D; Lehning et al., 2006) and make use of physiographic and climatological features in formulating the prior peak subgrid coefficient of variation distribution, perhaps even using climatological snow distribution patterns derived from high resolution fSCA imagery as pro-
posed in Sturm and Wagner (2010).

The posterior distributions were on the other hand much closer to the observed distributions for most of the years and sites considered. This suggests that there is sufficient information contained in the remotely sensed snow cover depletion to constrain the peak SSD estimates. On some occasions, especially for Bayelva 2015, the posterior SSD was far from the observed SSD
both in shape and in mean. However, the posterior estimate was still slightly better than the prior, indicating that the assimilation had a positive effect on the outcome. A similar marginal performance is found for Steinåen plateau in 2016, but the number of SWE point observations (see Table 1) is in this case not sufficient to reliably constrain the shape of the distribution, while peak mean SWE is slightly overestimated by the assimilation.

### 5.2  Evaluation of data assimilation schemes

Compared to the ES and PBS, which were used in previous studies (e.g. Durand et al., 2008; Girotto et al., 2014b; Margulis et al., 2015), ES-MDA exceeds or at least nearly matches the performance for all the evaluation metrics considered, i.e. bias, RMSE and correlation coefficient for fSCA, peak mean SWE and peak subgrid coefficient of variation. The performance gain over the ES is explained by the iterative nature of the ES-MDA, performing a sequence of smaller corrections in the analysis step as opposed to one abrupt correction (Emerick and Reynolds, 2013; Stordal and Elsheikh, 2015). Particularly in the case
of a non-linear model, as is the case for the SSM, this process of "simulated annealing" (c.f. Stordal and Elsheikh, 2015) in the ES-MDA leads to a better approximation of the posterior than a single analysis. In terms of error metrics, the RMSE with 100 ensemble members for the peak SWE ($\mu$) is lower for both the ES-MDA and the PBS than in the corresponding studies of Girotto et al. (2014b) and Margulis et al. (2015), although this could be attributed to the smaller size of the validation data set.




At least with a low number of ensemble-members the ES-MDA also outperforms the PBS: a possible reason for this is that the PBS posterior ensemble only spans the same range as the prior ensemble, and only changes the relative weights of the ensemble members in the analysis. Thus, if the prior ensemble is biased compared to the observations, it is unlikely that the analysis step in the PBS is capable of correcting the posterior towards the observations. In such a case, the region with high

likelihood is very small and not necessarily close to the observations. A good example is the 2008 season at Bayelva (c.f. Figure 4 and Figure 6) for which the prior is far away from the observed fSCA. Consequently, the PBS is unable to shift the ensemble outside the prior range as opposed to both the ES and the ES-MDA. In several years, the PBS also shows signs of degeneracy, i.e. a large part of the weight is carried by a very low number of particles. As the PBS is essentially a particle filter without re-sampling (Van Leeuwen, 2009), the weights can quickly converge on just a few particles in high-likelihood regions,

leaving the remaining particles redundant, even for low-dimensional systems with a relatively large number of observations and thus many analysis steps (as the one considered here). Note that both the ES and the ES-MDA are less prone to degeneracy (provided that the prior ensemble has sufficient spread) as they modify the position of the ensemble in the analysis step.

The sensitivity analysis for the ensemble size is consistent with higher dimensional models: the ES-MDA requires relatively

few ensemble members for a convergent performance, similar to the ensemble Kalman filter (EnKF; Evensen, 1994), while the PBS requires a relatively large ensemble size for convergence as with the particle filter (PF; see Van Leeuwen, 2009). Moreover, the number of assimilation cycles required for convergence of the ES-MDA, i.e. four cycles, is in line with previous studies (Emerick and Reynolds, 2013). While the PBS and the ES-MDA have not yet been compared with respect to improvement in RMSE for the three variables/parameters, our findings are in agreement with previous studies for both PBS vs ES (Margulis

et al., 2015) and ES-MDA vs ES (Emerick and Reynolds, 2013).

### 5.3 Computational expense

A major downside of the ES-MDA is the computational cost. The ES-MDA requires $N_a + 1$ ensemble model integrations and $N_a$ analysis steps, where $N_a$ (typically $\geq 2$) is the number of assimilation cycles (Section 3.3.2). The ES, on the other hand, requires only two ensemble model integrations and a single analysis step, while the PBS only requires a single ensemble

model integration and a single analysis step. In our case we used $N_a = 4$ so that the computational cost of the ES-MDA was significantly increased compared to the other schemes. For more complex models, such as Crocus (Vionnet et al., 2012) and SNOWPACK (Bartelt and Lehning, 2002), ES-MDA could thus prove prohibitively expensive. However, an adaptive version of the ES-MDA (Le et al., 2016) could be employed instead, in which the inflation coefficients are calculated on the fly based on a cost function and the iterations stop once the algorithm converges. This scheme could significantly reduce the computational

costs for applications of the ES-MDA, as it is would be equivalent to the ES in years for which the prior encompasses the fSCA retrievals, with only a single iteration. In years where the prior is far away from the observations to be assimilated, on the other hand, multiple data assimilation steps would be necessary. Furthermore, both the snow model (which does not interact with neighboring areas/grid cells) and the ES-MDA algorithm can be parallelized, so that the scheme could be applied to larger domains/many grid cells by using high performance computing.



## 5.4 Effects of observation error

As pointed out for Figure 6, all DA methods have problems constraining the spread of the peak subgrid coefficient of variation ($\chi$) although they can effectively pull the median in the right direction. A likely reason is the limited information available in the remotely sensed snow depletion, with either too sparse or too uncertain fSCA retrievals. Therefore, it could be worth considering fSCA retrievals from even more sensors, such as LandSat, PROBA-V and AVHRR, which increases the chances to obtain more cloud-free scenes. It is worth highlighting that $\chi$ is best constrained when a high number of accurate fSCA retrievals is available, such as Bayelva 2016 with Sentinel-2 retrievals added (c.f. Figure 4). Thus, with fSCA retrievals from different sensors added, it may be possible to better constrain the posterior $\chi$ ensemble, especially in the years where the SDC follows a well defined, monotonically decreasing pattern. Moreover, more accurate high resolution fSCA retrievals help in constraining the peak coefficient of variation: even with just a few additional high accuracy retrievals from Sentinel-2 the performance of the assimilation was markedly improved with respect to $\chi$ across all evaluation metrics. This clearly points towards the benefits of high resolution optical reflectance based fSCA retrievals from sensors on board the LandSat and Sentinel-2 satellites.

## 5.5 Outlook

Several extensions to the ensemble-based subgrid snow data assimilation discussed herein could be considered. The first is to change the grid scale of the assimilation framework from the order of $1\,\mathrm{km}$ by either going larger or smaller scale. In the case of the latter, it would be possible to just assimilate LandSat and Sentinel-2 based fSCA retrievals and operate at a grid scale on the order of $100\,\mathrm{m}$ in line with the work of Girotto et al. (2014a) and Margulis et al. (2016). For the former, one would aggregate the satellite retrievals yet further and perform the assimilation at a grid scale on the order of $10\,\mathrm{km}$ or larger. This implementation could be problematic as the probabilistic snow depletion curve formulation of Liston (2004) may no longer be justified since snow melt cannot be expected to be uniform across an entire grid cell.

Furthermore, the method could be applied to a larger domain in spatially distributed mode (i.e. multiple grid cells). In such a case, fSCA assimilation could be complemented by the assimilation of GRACE TWS and/or passive microwave (PM) SWE retrievals, which can also improve SWE estimates during the entire build-up, not only at peak SWE. Both TWS and PM could constrain the large-scale areal average SWE estimate within the domain and thus further bias-correct the multiple grid-scale peak mean SWE estimates. However, GRACE TWS retrievals feature a very coarse resolution (around $100\,\mathrm{km}$) so that they would only be useful in conjunction with fSCA retrievals for very large scale applications. On the other hand, the use of higher resolution PM SWE retrievals (order $25\,\mathrm{km}$) in the assimilation shows particular promise. Li et al. (2017) used the ES to assimilate PM SWE retrievals and estimate the SWE distribution at 90 m spatial resolution. De Lannoy et al. (2012) demonstrated assimilation of PM SWE in conjunction with fSCA retrievals in which the PM SWE retrievals are used to correct a global bias in all the SWE estimates that are conducted at the higher resolution of the fSCA retrievals. As discussed in Foster et al. (2005), the PM SWE retrievals are not accurate in complex topography and forested areas as well as for wet and deep snowpacks which





might limit the applicability of such multisensor assimilation approaches.

The major caveat to the assimilation of fSCA retrievals is the occurrence of clouds which causes extended gaps in time series obtained from optical sensors. As discussed, using fSCA retrievals from even more sensors could help to fill in the gaps in the
remotely sensed snow cover depletion and yet further constrain the peak subgrid coefficient of variation ($\chi$), which proved to be the most difficult parameter to constrain in the evaluation. In particular, the use of additional higher resolution fSCA retrievals, such as those from LandSat, with lower representativeness error (and thus observation error) could help in reducing the spread in $\chi$ and acquiring more reliable estimates.

To reduce the computational costs of the ES-MDA, the adaptive ES-MDA presented in Le et al. (2016) should be considered. In addition, implementing the bias corrected ES-MDA outlined in Stordal and Elsheikh (2015) may be worthwhile for future applications, especially when applied to a larger domain with possibly even larger misfits between the prior and the observations. Furthermore, implementing the scheme with a more complex snow model such as Crocus (Vionnet et al., 2012) or SNOWPACK (Bartelt and Lehning, 2002) may provide further benefits. For example, as noted by Durand et al. (2008), using
such multilayer snowpack models could open the possibility of assimilating surface snow grain size retrievals (c.f. Painter et al., 2009). Finally, employing a fully coupled land-atmosphere model could improve the performance even further. For example, the intermediate complexity atmospheric research model (ICAR; Gutmann et al., 2016) shows particular promise in terms of an atmospheric model that can efficiently and iteratively be run in ensemble mode, as required for applications of ES-MDA. In principle, one could run ICAR in ensemble mode coupled to a land surface model with an adequately complex snow scheme
and assimilate fSCA (and possibly PM and TWS) with the ES-MDA to deliver a snow reanalysis.

As snow is a crucial driver for many terrestrial and atmospheric processes, the ensemble-based subgrid snow data assimilation scheme may improve process modeling in a range of disciplines, in particular since the spatial resolution is considerably higher than in passive-microwave derived SWE data sets. For example, the subgrid variability of permafrost temperatures
is closely tied to that of SWE depth (e.g. Gisnås et al., 2016), which has major implications for permafrost mapping (e.g. Westermann et al., 2015b, 2016b). Similarly, snow cover information is an important component of many ecological models (e.g. Kohler and Aanes, 2004). Moreover, peak SWE is intimately linked to stream flow, and high-resolution information on the snow distribution hold significant potential for for hydrology and water resource management (Andreadis and Lettenmaier, 2006; Barnett et al., 2005). Finally, knowledge of snow distribution and snow melt is of interest for tourism given its importance
for e.g. skiing, hiking and backcountry travel.

## 6  Conclusions

In this study, we use an ensemble-based data assimilation scheme, the ensemble smoother with multiple data assimilation (ES-MDA), to estimate peak SWE distributions at the kilometer scale from time series of remotely sensed fSCA from MODIS



and Sentinel-2. The ES-MDA is combined with analytical Gaussian anamorphosis to update parameters that were either lower or double bounded in physical space. The data assimilation is applied to a simple snow model, based on the surface energy balance coupled to a probabilistic snow depletion curve. The scheme is driven by downscaled coarse ERA-Interim reanalysis data. As such, both the model forcing and the satellite retrievals are globally available.

The results are compared to in-situ data sets of snow distributions from high-arctic sites near Ny-Ålesund, Svalbard, so that the performance can be evaluated with respect to modeled fSCA, mean peak SWE and subgrid coefficient of variation. Thereby, the following conclusions can be drawn:

– At the kilometer scale, the ES-MDA is able to successfully assimilate fSCA retrievals into the simple snow model and
estimate the peak subgrid SWE distribution prior to the snow-melt.

– A physically-based interpolation of the remotely sensed fSCA time series is obtained that takes into account uncertainties in both the model and the retrievals.

– For peak mean SWE, the ES-MDA features a RMSE of 0.09 m w.e. which is lower than in previously published studies.

– For the peak subgrid coefficient of variation that controls the width and skewness of the distribution, the ES-MDA to a
certain extent fails to constrain the spread of the ensemble although the posterior median was usually pulled in the right direction.

– By including high resolution fSCA retrievals from Sentinel-2, the ensemble can be better constrained (in particular with respect to the coefficient of variation) which highlights the potential of additional high resolution fSCA retrievals from sensors on board the LandSat and Sentinel-2 satellites in future work.

– In line with previous studies, the ES-MDA converges with as low as 100 ensemble members and 4 assimilation cycles.

– With this configuration the fractional improvement in RMSE from prior to posterior is around 75%, 60% and 20% for the fSCA, peak mean SWE and peak subgrid coefficient of variation.

– The ES-MDA is shown to improve upon or at least nearly match the performance of the particle batch smoother and ensemble smoother for all evaluation metrics considered.

As the scheme exploits high and medium resolution satellite images from optical sensors, it is capable of estimating snow distribution at considerably higher spatial resolutions than traditional SWE products, e.g. based on passive microwave retrievals. On the other hand, the scheme can only recover the peak subgrid SWE distribution prior to the onset of melt, as opposed to providing information on the seasonal evolution of the snow distribution, so that it can rather complement than replace existing SWE retrieval algorithms. More generally, the method could become a part of satellite-era hydrometeorological reanalysis
schemes with a wide span of applications.



## 7 Data and code availability

Data and code are made available upon request from the corresponding author.

*Competing interests.* The authors declare that they have no conflict of interest.

*Acknowledgements.* This work was funded by SatPerm (project no. 239918; Research Council of Norway), in collaboration with EmblA

5 (project no. 56801; Nordforsk) and ESA GlobPermafrost (www.globpermafrost.info). Sebastian Westermann acknowledges additional support by COUP (project no. 2449037E10; JPI Climate; Research Council of Norway) and PermaNor (project no. 255331/E10; Research Council of Norway). Thomas V. Schuler acknowledges funding from ESCYMO (ref 224024; Research Council of Norway). This work is a contribution to the strategic research area LATICE at the University of Oslo.



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
