# Peer review of "Ensemble-based assimilation of fractional snow covered area satellite retrievals to estimate the snow distribution at Arctic sites"

_The Cryosphere, 2017_

## Referee Comment (RC1) · Anonymous Referee #1 · 1 Aug 2017

Ensemble-based assimilation of fractional snow covered area satellite retrievals to estimate snow distribution at a high Arctic site

The Cryosphere, Aalstad et al., 2017

The paper presents a case study in which MODIS (and Sentinel-2) fractional snow cover area retrievals are assimilated into a simple snow model to infer peak snow water equivalent and subgrid snow estimates. The ensemble smoother with multiple data assimilation technique is used. The paper adds interesting new insights to the discipline of snow data assimilation and could be considered for publication, after addressing the topics below:

1) - There is a weird mix of poor sentences in a very weak English language (too much to list, 1 random example: p.8, L5: "In addition, not accounted for... contribute to...") and parts with excellent English. My hope is that the excellent English parts are inserted by one of the co-authors and not copied from elsewhere. Please revise the entire paper for its language. Related to this, I would also suggest to revise the title to something like: "Ensemble-base assimilation of satellite-based fractional snow cover area to estimate the snow distribution at Arctic sites" (Arctic is already high-latitude; the sites do not have a particularly high elevation - I have no idea what "high" was referring to)

- Text: nice literature review, well done. The other text is a bit long in general, and has quite some repetitions in the discussion section in particular – please condense where possible. E.g. referring twice to the adaptive version of the ES-MDA, referring twice to the spatially distributed modeling, etc.

- what are "patterned gorund features"? A Nordic term?

2) MODIS, p.13, L.14: how exactly do you average the pixels for each study area? What to you mean by "all the pixels"? Would you include pixels with any (low) cloud fraction? (cloud fraction is given as additional information as a confidence measure for the fSCA retrieval estimates) What is the cloud cover limit? Mention explicitly that you are averaging the satellite data to the 1 km (I suspect), both for MODIS and Sentinel-2. Yet, it is mentioned that the Sentinel-2 data are averaged to the footprint of the snow surveys... so perhaps the latter is not right. If the resolution of the Sentinel-2 data, MODIS data and model are different, please explain how you reconcile the space-mismatch. Also mention the 1-km spatial resolution upfront in the modeling for clarity.

3) Ensemble data assimilation:

P15, L4 explicitly name the "parameters" as "perturbation parameters" to avoid confusion with model parameters.

P16, L8-9: "remaining parameters", "prior parameter ensemble": again refine the word choice here or define "state", "parameter" and "perturbations" more clearly or efficiently upfront, because this sentence is referring to two types of parameters and confusing.

P16, L18: identify here (again) what is in the state and parameter vector. I suspect that there are no "snow model parameters" included anywhere, but it would be good to explicitly mention this. Clarify upfront that perturbation parameters are updated and not state variables or model parameters, as is often the case in other hydrology/cryosphere research.

Fig.3: replace block with "Update states" → "Propagate state" (I believe that the updating is done on the parameters in the ES-MDA Analysis step)

4) Have you evaluated the updated parameters (result of Eq. 21) themselves? Do the bias estimates make sense? E.g. can you compare the re-analysis forcings, the bias-corrected ones and in situ observed meteorological data?

5) P.20, L17 and Fig 4: peak measurements. . . who or what knows that this measurement is taken at the peak? How can we know for sure that it is a peak measurement, if there is only one data point? Why is that peak measurement always located on May 1st in Fig 4? The peak measurement must be at a different time every year. . .

6) Fig 5: these distributions seem not to be cross-masked, i.e. why are the prior and posterior estimates not cross-masked to the times and locations of the in situ observations?

7) Table 4, Figure 7: clarify in the caption which validation data are used, where and during which period.

8) P.23, L6: in situ fSCA retrievals – are these the camera-based data? Please clarify. It would also help to use the same term to refer to validation data throughout the paper – i.e. in situ vs ground-based vs ground truth vs "Field measurements" (section 2.2)

9) P24, 15: "lowest improvements": how confident are you with this statement? With

only 8 samples, it is hard to get statistical significance of any sort.

10) Table 5: not sure if this exercise has any value with only 3 observations as validation... what is the confidence interval on these metrics?

11) Section 4.3: "Effects of observation error and assimilation frequency" ?

12) Conclusions, P31: "For peak mean SWE... lower than in previously...": this is an apples to oranges comparison to published work over different regions and different periods and thus invalid; please remove.
* * *

---

## Referee Comment (RC2) · Anonymous Referee #2 · 21 Aug 2017

Ensemble-based assimilation of fractional snow covered area satellite retrievals to estimate snow distribution at a high Arctic site

The Cryosphere, Aalstad et al., 2017

This paper shows an analysis of the results of three different assimilation algorithms applied to snow variables (in particular SWE, fSCA and sub-grid coefficient of variation) over an Arctic study site. The assimilation algorithms used are the Ensemble Smoother (ES), Particle Batch Smoother (PBS) and a newly introduced Ensemble Smoother Multiple Data Assimilation (ES-MDA) technique. The results show significant improvements in all evaluation metrics for the ES-MDA technique, matching or

improving the results obtained using the ES and PBS. The ES-MDA is more robust as it avoids degeneracy and other problems of the other two techniques however this comes at an expended computational cost.

The paper is well written (albeit it needs more work in terms of grammar/phrasing, I recommend one final review by native-english speaker) and very clear. The methodology section might be improved by including examples of the method using figures. As it is right now is very mathematical, which is fine but reduces the possibility of understanding the workings of the method by other researchers on the field. The literature review is very comprehensive and might be improved if condensed. The paper further illustrates the extreme utility of data assimilation frameworks in the context of snow process estimation and I recommend it for publication after minor revisions.

Specific comments:

2-3: The amount of smoothing depends on the type of terrain - wouldn't expect this effect to be significant beyond smoothing microtopography (i.e., 1-2 m vertical scale). 13: Probably only precipitation and wind are space-time variant, topography and vegetation shouldn't be considered as dynamic. Radiation is also space-time variant however the direct component climatology might be relatively invariant every year - though I would expect that for high latitudes this is not necessarily true,

17: Maybe it is worthy citing Cortés et al. (2016; 2017) for a more direct comparison with PBS metrics derived over similar study regions. Both papers include similar validation data (snow surveys), while Margulis et al is focused on point-data (stations).

31: Define undulating. 32: Typo (ground)

7: Please clarify what do you mean by this term? Also - clarify what "external" processes are not considered (wind redistribution?)

26: Is there a range defined for this parameter?

10: Is the daily time step a result from aggregating internal hourly calculations?

4-5: It would be useful to include a quantification of how many images were available per assimilation season for each site. How were clouds identified and masked out?

7: Curious if the use of independent multiplicative biases for accumulation and melt would result in inconsistent accumulations? (For example b*M>b*P?) 8: When you mention constant multiplicative biases - does this mean the bias is unaltered throughout the year? 13: The PBS requires that the ensemble includes the observation, thus if no bias is assigned a priori then the PBS might not be applicable as some degree of bias correction is needed.

6: The reduction in spread is a direct consequence of any assimilation algorithm, it would be more useful to assess if the constrain in uncertainty of the posterior is consistent with the observations (i.e., are you underestimating uncertainty after assimilation?)

A scatterplot would be useful to compare the posterior results for all methods. Including

the stats is correct but scatterplot allows for more context.

Table 5: A perfect correlation of 1.0 was obtained? Would be useful to have the scatterplots in order to inform the reader with more details on the results.

Page 27 32: It is very difficult to compare RMSE across studies due to the differences in methodology/data. I would stick to the comparison performed within the paper as it allows for more controlled conditions.

Page 28 3-4: More than biased, if the prior ensemble doesn't cover the observations then the PBS would be unable to replicate the observation. Bias in the ensemble per se is not a problem for the PBS. The comparison between PBS/ES from previous papers with the current method is not as straightforward, asides from the obvious differences in regions there are differences in validation methodology and particularly in the number of fSCA measurements assimilated. Landsat fSCA assimilation results in 10-15 observations per year, while MODIS probably results in an order of magnitude greater.

References:

Cortés, G., Girotto, M., & Margulis, S. (2016). Snow process estimation over the extratropical Andes using a data assimilation framework integrating MERRA data and Landsat imagery. Water Resources Research, 52(4), 2582-2600.

Cortés, G., & Margulis, S. (2017). Impacts of El Niño and La Niña on interannual snow accumulation in the Andes: results from a high‐resolution 31‐year reanalysis. Geophysical Research Letters.
* * *

---

## Author Comment (AC1) · 24 Oct 2017

**Reply to Reviewer 1**

We are grateful to the reviewer for the thoughtful comments and suggestions to our manuscript. We have compiled a revised version and in the following provide a point-by-point reply to all issues raised. The reviewer's comments appear in bold font and our replies in normal font. Excerpts from and changes to the manuscript are quoted in italics. Page and line numbers refer to positions in the original manuscript.

[Figure]

**Ensemble-based assimilation of fractional snow covered area satellite retrievals to estimate snow distribution at a high Arctic site**

**The Cryosphere, Aalstad et al., 2017**

**The paper presents a case study in which MODIS (and Sentinel-2) fractional snow cover area retrievals are assimilated into a simple snow model to infer peak snow water equivalent and subgrid snow estimates. The ensemble smoother with multiple data assimilation technique is used. The paper adds interesting new insights to the discipline of snow data assimilation and could be considered for publication, after addressing the topics below:**

**1) - There is a weird mix of poor sentences in a very weak English language (too much to list, 1 random example: p.8, L5: "In addition, not accounted for. . . contribute to. . .") and parts with excellent English. My hope is that the excellent English parts are inserted by one of the co-authors and not copied from elsewhere. Please revise the entire paper for its language. Related to this, I would also suggest to revise the title to something like: "Ensemble-base assimilation of satellite-based fractional snow cover area to estimate the snow distribution at Arctic sites" (Arctic is already high-latitude; the sites do not have a particularly high elevation - I have no idea what "high" was referring to)**

Following the reviewer's suggestion the entire manuscript has been revised for its language.

According to the reviewer's suggestion, the title has been revised to
*"Ensemble-based assimilation of fractional snow covered area satellite retrievals to estimate the snow distribution at Arctic sites"*

**- Text: nice literature review, well done. The other text is a bit long in general, and has quite some repetitions in the discussion section in particular – please condense where possible. E.g. referring twice to the adaptive version of the ES-MDA, referring twice to the spatially distributed modeling, etc.**

The potential for spatially distributed modeling and assimilation is now only mentioned in the outlook (Section 5.4). Furthermore, the adaptive ES-MDA is now first mentioned in Section 5.2 (Evaluation of data assimilation schemes) and once more briefly in the outlook. In addition, we have removed the statements concerning the novelty and basis of our work early in Section 3.3 as this was already mentioned in the introduction. Some necessary repetition between the outlook (Section 5.5) and other parts of the discussion remains, but this is kept as brief as possible.

**- what are "patterned gorund features"? A Nordic term?**

Thanks for spotting a typo. The term has been changed to

*"patterned ground features"*

which refers to a phenomenon in periglacial regions where patterns, such as sorted circles, form in the ground material.

**2) MODIS, p.13, L.14: how exactly do you average the pixels for each study area? What to you mean by "all the pixels"? Would you include pixels with any (low) cloud fraction? (cloud fraction is given as additional information as a confidence measure for the fSCA retrieval estimates) What is the cloud cover limit? Mention explicitly that you are averaging the satellite data to the 1 km (I suspect), both for MODIS and Sentinel-2. Yet, it is mentioned that the Sentinel-2 data are averaged to the footprint of the snow surveys. . . so perhaps the latter is not right. If the resolution of the Sentinel-2 data, MODIS data and model are different, please explain how you reconcile the space-mismatch. Also mention the 1-km spatial resolution upfront in the modeling for clarity.**

For each study site and day we simply take the mean fSCA over all the corresponding MODIS pixels shown in Figure 1. This average is only taken if none of the pixels are flagged as cloudy by the MODIS cloud mask. The MOD10A1 and MYD10A1 version 6 products only accept pixels flagged as cloud free (i.e. either "confident clear", "probably clear" or "uncertain clear") by the MODIS cloud mask (see Riggs and Hall, 2016). Cloud fraction is not given as a confidence measure in these products (Riggs and Hall, 2016). If cloud free pixels are available from both Terra and Aqua, then Terra pixels are chosen. To clarify how clouds are dealt with we have added the following sentence to Section 3.2.1:

*'We average over all the pixels for each day and study site (see Figure 1). This average is only taken if cloud free (as determined by the MODIS cloud mask) retrievals are available for each of these pixels."*

As stated in the Section 3.2.2, the Sentinel-2 fSCA retrievals are mapped to the approximate footprint of (i.e. the area encompassed by) the snow surveys. The areal extent of the Sentinel-2 retrievals is close to those given in Table 1. As such, the

following sentence has been added to Section 3.2.2:

*"Therefore, the areal extent of the Sentinel-2 fSCA retrievals closely matches the areas of the corresponding study sites given in Table 1."*

Consequently, there is a space-mismatch between the MODIS and Sentinel-2 fSCA retrievals. The latter have lower representativeness error as they provide a better match to the area covered by the snow surveys. The point of the exercise in Section 4.3 is to check whether or not this mismatch has a considerable effect on the assimilation results. Still, the space-mismatch in the MODIS pixels is not huge (Figure 1). Moreover, the space-mismatch is reconciled by the difference in the observation error variance (RMSE$^2$) between the MODIS and Sentinel-2 retrievals, determined based on the field measurements of fSCA, where the Sentinel-2 error is considerably lower.

As for the resolution of the modeling, the following has been added to the end of the modeling section (Section 3.1.2) for clarification:

*"The model resolution is defined by the footprint of (area encompassed by) the snow surveys for each site (see Table 1 and Figure 1)."*

In addition, the following has been added to the end of the forcing section (Section 3.1.3):

*"While the resolution of the downscaled forcing data do not exactly match the model resolution (i.e. the footprint of the snow surveys, Section 3.1.2), the mismatch is small considering the gentle topography of the study sites (Section 2.1)."*

**3) Ensemble data assimilation:**

**P15, L4 explicitly name the "parameters" as "perturbation parameters" to avoid confusion with model parameters.**

Done.

**P16, L8-9: "remaining parameters", "prior parameter ensemble": again refine the word choice here or define "state", "parameter" and "perturbations" more clearly or efficiently upfront, because this sentence is referring to two types of parameters and confusing.**

Since peak mean SWE is also treated as a state variable in the SSM, the sentence has been reformulated to

*"We emphasize that through the perturbation parameters we effectively perturb the melt rate, precipitation rate and coefficient of variation. By performing a subsequent ensemble integration of the SSM we also get an ensemble of state variables that are consistent with the prior perturbation parameter ensemble."*

Where we have also explicitly referred to the parameters that are perturbed as perturbation parameters. In addition, we have changed *"prior ensemble of parameters"* in Table 4 to *"prior ensemble of perturbation parameters"*.

**P16, L18: identify here (again) what is in the state and parameter vector. I suspect that there are no "snow model parameters" included anywhere, but it would be good to explicitly mention this. Clarify upfront that perturbation parameters are updated and not state variables or model parameters, as is often the case in other hydrology/cryosphere research.**

The section starting from the middle of P16 L16 running to P16 L21 has been modified to

*"Let $N_e$, $N_o$, $N_a$, $N_s$, $N_p$ and $N_t$ denote the number of ensemble members, observations, assimilation cycles, state variables, perturbed parameters and time steps during an annual (September-August) model integration. $\mathbf{X}$ is the $(N_s \times N_t) \times N_e$ matrix containing the ensemble of states ($fSCA_{n,j}$, $D_{m,n,j}$, $\overline{D}_{n,j}$ and $\mu_{n,j}$) and $\boldsymbol{\Theta}$ is the $N_p \times N_e$ matrix containing the ensemble of perturbation parameters listed in Table 4. The $N_o \times 1$ observation vector $\mathbf{y}$ contains all the fSCA satellite retrievals during the ablation season (Section 3.2), $\mathbf{Y}$ is the $N_o \times N_e$ matrix containing the ensemble of perturbed fSCA satellite retrievals and $\widehat{\mathbf{Y}}$ is the $N_o \times N_e$ matrix containing the ensemble of predicted fSCA observations. Additionally, $\mathbf{H}$ is the observation operator (mapping from the state space to the observation space) and $\mathbf{R}$ is the $N_o \times N_o$ observation error covariance matrix which is a diagonal matrix containing the observation error variances (Section 3.2)."*

To emphasize that the model constants (model parameters) in Table 2 are fixed and that only the perturbation parameters are updated in the analysis we have changed the sentence on P18 L9 to

*"We emphasize that the analysis step (21) only updates the perturbation parameters*

*and a consistent ensemble of states is found from the subsequent ensemble model integration. The model constants listed in Table 2 remain unchanged by the analysis and the integration.".*

In addition, all further mentions of *"parameters"* have been corrected to *"perturbation parameters"*.

**Fig.3: replace block with "Update states" ⟶ "Propagate state" (I believe that the updating is done on the parameters in the ES-MDA Analysis step)**

Done. For consistency, in this Figure the block *"Generate Prior Parameter Ensemble'"* was changed to *"Generate prior perturbation parameter ensemble"*.

**4) Have you evaluated the updated parameters (result of Eq. 21) themselves? Do the bias estimates make sense? E.g. can you compare the re-analysis forcings, the bias-corrected ones and in situ observed meteorological data?**Ăă

Snowfall observations are not available for our field sites (Boike et al., 2017) and are very difficult to conduct in the Arctic due to undercatch (e.g. Førland and Hanssen-Bauer, 2000). For the snowmelt, a field based estimate was available from Westermann et al. (2009). We have added the following to the end of Section 4.1:

*"The posterior bias parameters can be directly evaluated by comparing the bias corrected forcing to field measurements. Due to a lack of snowfall observations (see Boike et al., 2017), an evaluation of the precipitation bias parameter is not possible. However, the melt bias parameter can be evaluated by comparing the snowmelt flux*

*(which is directly proportional to the perturbed melt depth) to field-based values. For June 2008, Westermann et al. (2009) estimate an average snowmelt flux of 27 Wm$^{-2}$ , which compares well to the ES-MDA posterior median estimate (averaged for the same period) of 29 Wm-2 , while the prior median estimate is too low with 19 Wm-2."*

**5) P.20, L17 and Fig 4: peak measurements. . . who or what knows that this measurement is taken at the peak? How can we know for sure that it is a peak measurement, if there is only one data point? Why is that peak measurement always located on May 1st in Fig 4? The peak measurement must be at a different time every year. . .**

Only a single survey is available for each site for a given snow season. The times of these surveys (April/May), coincide closely with the peak SWE. We have added a quantification of the associated error using snow depth measurements from a sonic ranger at the Bayelva climate station to Section 2.2 (P8 L7):

*"Although the snow surveys coincide closely with peak SWE, some accumulation (ablation) may occur after (before) the surveys. To assess the magnitude of this error source, we used snow depth measurements at the Bayelva station (Boike et al., 2017) to compare the snow depth at the survey dates to the maximum snow depth for each snow season. We found an average relative difference of 8% (maximum: 17%, minimum: 0.3%)."*

Figure 4 has also been changed. It now shows the measured peak mean SWE $\mu$ (as determined by the snow surveys) over the whole snow season as a horizontal dotted black line. The caption of Figure 4 was changed to

*"Time evolution of the prior (red) and ES-MDA ($N_a = 4$, $N_e = 10^2$) posterior (blue) fSCA (first column) and mean SWE ($\overline{D}$; second column); shading: $90^{th}$ percentile range; solid lines: ensemble median; yellow dots: assimilated MODIS and Sentinel-2 fSCA retrievals; dotted black line: independently observed peak mean SWE ($\mu$) from snow surveys (Section 2.2); $x$-axis: months. These results are from a single run."*

**6) Fig 5: these distributions seem not to be cross-masked, i.e. why are the prior and posterior estimates not cross-masked to the times and locations of the in situ observations?**

We are not sure what the reviewer means by cross-masked. As discussed above, the times of the snow surveys coincide closely with peak SWE. In Figure 5 we are comparing the prior and posterior estimates of the peak subgrid SWE distributions with the corresponding empirical distribution measured in the field through the snow surveys for each study site and snow season. To clarify this, the caption in Figure 5 has been changed to

*"Prior (red), ES-MDA ($N_a = 4$, $N_e = 10^2$) posterior (blue) and the corresponding independently observed (from snow surveys; dashed black) peak subgrid SWE distributions; shaded areas: $90^{th}$ percentile range; solid lines: ensemble median; markers: mean value. These results are from a single run."*

**7) Table 4, Figure 7: clarify in the caption which validation data are used, where and during which period.**

The caption of Figure 8 (previously Figure 7) was changed to

*"Fractional improvement in RMSE through the analysis step ($1$ being perfect and $0$ no effect) as a function of the number of ensemble members for the fSCA, peak mean SWE $\mu$ and coefficient of variation $\chi$; top left: particle batch smoother, PBS; top right; ensemble smoother, ES; bottom left; ensemble smoother with multiple data assimilation, ES-MDA; bottom right: FI as a function of assimilation cycles in the ES-MDA. The FI for $N_e \leq 100$ are averaged over 100 independent ensemble model integrations. Errors were computed based on comparisons to all the corresponding field measurements presented in Section 2.2."*

The caption of Table 5 (previously Table 4) was changed to

*"Summary of evaluation metrics, i.e. bias, RMSE and square correlation coefficient ($R^2$), for the fSCA, peak SWE ($\mu$) and peak subgrid coefficient of variation ($\chi$). These metrics are based on comparisons to all the field measurements presented in Section 2.2 with the number of observations for the comparisons in brackets next to the corresponding symbols. All the metrics are averaged over 100 independent runs each with 100 ensemble members. The ES-MDA was run with $N_a = 4$ assimilation cycles."*

For consistency, the caption in Table 6 (previously Table 5) was changed to

*"Summary of evaluation metrics, i.e. fractional improvement in bias and RMSE as well as prior and posterior square correlation coefficient ($R^2$), using the ES-MDA ($N_e = 10^2$, $N_a = 4$) for peak mean SWE ($\mu$) and coefficient of variation ($\chi$) when assimilating only MODIS as well as assimilating both MODIS and Sentinel-2 observations. These metrics are based on a comparison to all the snow surveys conducted in 2016 (see Table 1) and are averaged over 100 independent runs each with 100 ensemble members."*

**8) P.23, L6: in situ fSCA retrievals – are these the camera-based data? Please clarify. It would also help to use the same term to refer to validation data throughout the paper – i.e. in situ vs ground-based vs ground truth vs "Field measurements" (section 2.2)**

Yes, these are the fSCA field measurements based on data from the automatic camera system, UAV and GPS-based surveys discussed in Section 2.2. To clarify and avoid confusion with the satellite retrievals, the corresponding section of P23 L6 was modified to

*"field measurements of fSCA (Section 2.2.)"*

For consistency, all mentions of *"in situ"* (with one exception) or *"ground based"* or *"ground truth"* (with one exception) have been changed to *"field measurements"*.

**9) P24, 15: "lowest improvements": how confident are you with this statement? With only 8 samples, it is hard to get statistical significance of any sort.**

To qualify this statement, the sentence on P24 L15 was changed to

*"For all schemes the available validation data suggests that the greatest improvements are achieved for fSCA, followed by peak mean SWE, while by far the lowest improvements are found for the peak subgrid coefficient of variation."*

**10) Table 5: not sure if this exercise has any value with only 3 observations as validation. . . what is the confidence interval on these metrics?**

As the Sentinel-2 mission is relatively new (the first satellite, Sentinel-2A, was launched in June 2015), only 3 snow surveys (from 2016) are available for comparison at our study sites since the start of the mission. As such we have added the following cautionary statement to Section 4.3 (P26 L15):

*"We emphasize that this evaluation is based on the only 3 available field measurements of $\mu$ and $\chi$ in 2016 (from the snow surveys), so that these preliminary results need to be consolidated by future studies with more validation data."*

**11) Section 4.3: "Effects of observation error and assimilation frequency" ?**

The heading has been changed accordingly. The same change was made to the heading of Section 5.3 (previously Section 5.4).

**12) Conclusions, P31: "For peak mean SWE. . . lower than in previously. . .": this is an apples to oranges comparison to published work over different regions and different periods and thus invalid; please remove.**

Done. The end of the sentence has been removed so that it now reads

*"For the peak mean SWE, the ES-MDA features an average RMSE of 0.09 m w.e. compared to field measurements"*

Thank you once again for all the helpful comments and suggestions,
On behalf of all the co-authors,
Kristoffer Aalstad

**References**

Boike, J., Juszak, I., Lange, S., Chadburn, S., Burke, E., Overduin, P., Roth, K., Ippisch, O., Bornemann, N., Stern, L., Gouttevin, I., Hauber, E., and Westermann, S. (2017). A 20-year record (1998-2017) of permafrost, active layer, and meteorological conditions at a High Arctic permafrost research site (Bayelva, Spitsbergen): an opportunity to validate remote sensing data and land surface, snow and permafrost models. *Earth System Science Data Discussions*, pages 1–86. 10.5194/essd-2017-100.
Førland, J. E. and Hanssen-Bauer, I. (2000). Increased Precipitation in the Norwegian Arctic: True or False? *Climatic Change*, 46:485–509. 10.1023/A:1005613304674.
Riggs, G. A. and Hall, D. K. (2016). MODIS Snow Products Collection 6 User Guide. Version 1.0.
Westermann, S., Lüers, J., Langer, M., Piel, K., and Boike, J. (2009). The annual surface energy budget of a high-arctic permafrost site on Svalbard, Norway. *The Cryosphere*, 3:245–263. 10.5194/tc-3-245-2009.

---

## Author Comment (AC2) · 24 Oct 2017

**Reply to Reviewer 2**

We are grateful to the reviewer for the thoughtful comments and suggestions to our manuscript. We have compiled a revised version and in the following provide a point-by-point reply to all issues raised. The reviewer's comments appear in bold font and our replies in normal font. Excerpts from and changes to the manuscript are quoted in italics. Page and line numbers refer to positions in the original manuscript.

**Ensemble-based assimilation of fractional snow covered area satellite retrievals to estimate snow distribution at a high Arctic site**

**The Cryosphere, Aalstad et al., 2017**

This paper shows an analysis of the results of three different assimilation algorithms applied to snow variables (in particular SWE, fSCA and sub-grid coefficient of variation) over an Arctic study site. The assimilation algorithms used are the Ensemble Smoother (ES), Particle Batch Smoother (PBS) and a newly introduced Ensemble Smoother Multiple Data Assimilation (ES-MDA) technique. The results show significant improvements in all evaluation metrics for the ES-MDA technique, matching or improving the results obtained using the ES and PBS. The ES-MDA is more robust as it avoids degeneracy and other problems of the other two techniques however this comes at an expended computational cost.

The paper is well written (albeit it needs more work in terms of grammar/phrasing, I recommend one final review by native-english speaker) and very clear. The methodology section might be improved by including examples of the method using figures. As it is right now is very mathematical, which is fine but reduces the possibility of understanding the workings of the method by other researchers on the field. The literature review is very comprehensive and might be improved if condensed. The paper further illustrates the extreme utility of data assimilation frameworks in the context of snow process estimation and I recommend it for publication after minor revisions.

Following the reviewer's suggestion the entire paper has been revised for its grammar/phrasing.

We agree with the reviewer that our presentation is quite mathematical. In a sense, this is unavoidable given the mathematical nature of data assimilation. With Figure 3 we provide a more schematic overview of the work flow in the methodology. We find that even after the revisions the manuscript is still quite long and thus chose to avoid adding additional figures.

The literature review has been condensed by removing many of the fine details regarding the results of different studies. We also refrained from stating the spatial resolution of the cited studies.

**Specific comments:**

**Page 2**

**2-3: The amount of smoothing depends on the type of terrain - wouldn't expect this effect to be significant beyond smoothing microtopography (i.e., 1-2 m vertical scale).**

We have removed this sentence.

**13: Probably only precipitation and wind are space-time variant, topography and vegetation shouldn't be considered as dynamic. Radiation is also space-time variant however the direct component climatology might be relatively invariant every year - though I would expect that for high latitudes this is not necessarily**

**true,**

Yes, topography and vegetation are relatively fixed. In clear conditions the direct shortwave component of the radiation is mainly fixed by solar geometry and the local topography and is as such relatively invariant from year to year in the absence of clouds. Still, clouds are common in the Arctic and along with variations in the surface albedo this can make net radiation a highly dynamic variable. To clarify, the sentences starting on P2 L12 and running up to the start of P2 L15 have been reformulated to

*"The primary controls on the distribution and variability of SWE are topography, vegetation, precipitation, wind, radiation and avalanching (Sturm and Wagner, 2010; Clark et al., 2011). While topography and vegetation are relatively fixed in time, the other controls vary strongly over a range of spatiotemporal scales."*

**Page 4**

**17: Maybe it is worthy citing Cortés et al. (2016; 2017) for a more direct comparison with PBS metrics derived over similar study regions. Both papers include similar validation data (snow surveys), while Margulis et al is focused on point-data (stations).**

We agree that both Cortés et al. (2016) and Cortés and Margulis (2017) are valuable references for probabilistic SWE reconstruction in sparsely instrumented regions and have thus added the following to P4 L19

*"Cortés et al. (2016) applied the same PBS framework to construct a 30 year reanalysis of SWE over 6 instrumented basins in the Andes. Cortés and Margulis (2017)*

*subsequently adopted this approach to perform a 31 year SWE reanalysis over the
entire extratropical Andes."*

**Page 5**

**31: Define undulating.**

By undulating we mean gentle topography with small hills. The sentence was changed
to

*"All sites feature gently undulating topography with small hills and surfaces character-
ized by patterned ground features, leading to strong differences in snow cover due to
wind drift."*

**32: Typo (ground)**

Thanks for spotting this typo. It has been corrected.

**Page 9**

**7: Please clarify what do you mean by this term? Also - clarify what "external"
processes are not considered (wind redistribution?)**

By this we mean any process occurring inside the snowpack itself. Since refreezing
is treated at sub-daily resolution and metamorphism is treated implicitly by the snow

albedo parametrization we have reformulated the sentence to

*"Many internal snow processes (occurring inside the snowpack), including heat conduction and melt water percolation, are omitted. In addition, several external processes such as sublimation and deposition are ignored."*

Wind redistribution is treated implicitly by the probabilistic SDC of Liston (2004) through a non-zero peak coefficient of variation ($\chi$) that accounts for a non-uniform peak subgrid SWE distribution (SSD). The shape of this peak SSD (and thus $\chi$) will be partly controlled by wind redistribution during the accumulation season.

**Page 10**

**26: Is there a range defined for this parameter?**

Yes, in Table 4 (previously Table 3). A reference to the table was added to the text:

*"$Q_0$ is a perturbation parameter (see Table 4) that is updated in the assimilation,"*

**Page 11**

**10: Is the daily time step a result from aggregating internal hourly calculations?**

Yes, the forcing is aggregated from subdaily to daily resolution as discussed in Section 3.1.3.

**Page 13**

**4-5: It would be useful to include a quantification of how many images were available per assimilation season for each site. How were clouds identified and masked out?**

A new Table (Table 3) has been added that lists the number of available scenes per melt season for both MODIS and Sentinel-2 for each study site.

For MODIS, clouds were masked out automatically by the MODIS cloud mask (see Riggs and Hall, 2016). The following sentence has been added to Section 3.2.1:

*"We average over all the pixels for each day and study site (see Figure 1). This average is only taken if cloud free (as determined by the MODIS cloud mask) retrievals are available for each of these pixels."*

For Sentinel-2, clouds were masked out manually in the scene selection as specified in Section 3.2.2.

**Page 15**

**7: Curious if the use of independent multiplicative biases for accumulation and melt would result in inconsistent accumulations? (For example b*M>b*P?)**

This is an interesting point. We have added the following to the discussion (Section 5.1):

*"An inherent equifinality problem (see Beven, 2006) exists in SWE reconstruction since different perturbation parameter sets can provide similar results. For example, if the prior fSCA melts out earlier than the observations this could be due to the prior precipitation having a negative bias, the prior melt having a positive bias or a combination of these two. The opposite would be true if the prior fSCA melts out too late. It is not possible to resolve this equifinality problem with observations of fSCA alone. A key assumption in deterministic SWE reconstruction is that the melt flux is more constrained than the precipitation so that uncertainty in the melt is ignored (Slater et al., 2013). We perturb both the precipitation and the melt, although the latter is assigned a lower uncertainty (Table 4). Through the assimilation we obtain snowmelts that are consistent with the observed snow cover depletion. The close match of the posterior peak mean SWE estimates to the independent field measurements (Figure 7) suggests that the assimilation yields consistent accumulations and that the inherent equifinality problem is of minor consequence."*

**8: When you mention constant multiplicative biases - does this mean the bias is unaltered throughout the year?**

Yes, this has been clarified in the text where we have changed *"constant multiplicative biases"* to *"constant multiplicative biases (fixed throughout the annual integration)".*

**13: The PBS requires that the ensemble includes the observation, thus if no bias is assigned a priori then the PBS might not be applicable as some degree of bias correction is needed.**

Both of the prior bias parameters are modeled as lognormal random variables with unit mean but a non zero variance. So a considerable bias is assigned a priori for some of the ensemble members. We agree that the PBS requires that the prior ensemble encompasses the assimilated observations and we view this as a weakness. For example in 2008 for Bayelva the prior fSCA ensemble is positively biased and does does not encompass the observations, so the PBS performs poorer than the ES-based schemes. We could have changed the prior mean of the bias parameters for this particular year but decided not to. In the application of Bayesian data assimilation the prior should always be set without knowledge of the observations that are considered in the likelihood otherwise it is by definition not a prior. See also the reply to the comment concerning Page 28 L3-4.

**Page 20**

**6: The reduction in spread is a direct consequence of any assimilation algorithm, it would be more useful to assess if the constrain in uncertainty of the posterior is consistent with the observations (i.e., are you underestimating uncertainty after assimilation?)**

We agree with the reviewer. As such, the following paragraph was added to the end of Section 4.1:

*"In ensemble-based data assimilation the spread of the posterior ensemble should represent the uncertainty. To verify this one can compare two metrics: the residual, i.e. the instantaneous posterior RMSE of the ensemble relative to the corresponding independent field measurement, and the ensemble standard deviation (e.g. Evensen,*

*2009). For this comparison we define the relative residual as the ratio of the residual to the standard deviation. Ideally this ratio should have a value of 1 which indicates that the two metrics are equal so that the posterior ensemble spread accurately captures the estimation uncertainty. For the fSCA, peak mean SWE and peak subgrid coefficient of variation the average (over all available field measurements) relative residuals were 2.22, 1.53 and 1.66 respectively, so the posterior ensemble underestimates the uncertainty. This effect has been extensively described by Evensen (2009), it arises in part because of model structural errors related to neglected physical processes (Section 3.1). Still, the assimilation is generally able to simultaneously (but not to the same extent) reduce the spread and the error in the ensemble (Figure 4). "*

**Page 21**

**A scatterplot would be useful to compare the posterior results for all methods. Including the stats is correct but scatterplot allows for more context.**

We agree and have included a scatter plot for a single run of all three schemes and the prior. This is included for orientation as a new Figure 6. The following line was included at the end of P23 L10:

*"The scatter plots in Figure 6 visualize the performance of the prior and all the considered DA schemes relative to the field measurements."*

**Page 26**

**Table 5: A perfect correlation of 1.0 was obtained? Would be useful to have the**

[Figure]

**scatterplots in order to inform the reader with more details on the results.**

This perfect correlation is based on a comparisson to just 3 observations. Because of the low number of observations a scatter plot is not informative and would unnecessarily add to the length of the manuscript. We have added the following cautionary statement regarding the limited number of observations used in this evaluation to Section 4.3 (P26 L9):

*"We emphasize that this evaluation is based on the only 3 available field measurements of $\mu$ and $\chi$ in 2016 (from the snow surveys), so that these preliminary results need to be consolidated by future studies with more validation data."*

**Page 27**

**32: It is very difficult to compare RMSE across studies due to the differences in methodology/data. I would stick to the comparison performed within the paper as it allows for more controlled conditions.**

We agree and have removed this comparison.

**Page 28**

**3-4: More than biased, if the prior ensemble doesn't cover the observations then the PBS would be unable to replicate the observation. Bias in the ensemble per se is not a problem for the PBS. The comparison between PBS/ES from previous papers with the current method is not as straightforward, asides from the**

**obvious differences in regions there are differences in validation methodology and particularly in the number of fSCA measurements assimilated. Landsat fSCA assimilation results in 10-15 observations per year, while MODIS probably results in an order of magnitude greater.**

To clarify, we have changed this sentence to:

*"Thus, if the prior ensemble is so biased that it does not encompass the observations, the PBS is incapable of correcting the posterior towards the observations outside the bounds of the prior."*

For the next part of the comment we assume that the reviewer is referring to the sensitivity analysis around the middle of P28 L18. All other comparisons to the results of previous probabilistic reconstruction schemes have been removed. Here we are just comparing the relative performance of the ES to the PBS. Of course the locations and assimilated data sets are different, with MODIS definitely having a higher temporal coverage. Still, it is positive to see that the results achieved from previous studies that the PBS generally outperforms the ES matches our own findings and so we do not see why this should not be included. We do not say that our studies are the same but simply that the results agree. The same applies to the study of Emerick and Reynolds (2013), in a completely different field, we still expect the same kind of relative performance for the data assimilation schemes in a sensitivity analysis with a non-linear model which is indeed what we find.

Thank you once again for all the helpful comments and suggestions,
On behalf of all the co-authors,
Kristoffer Aalstad

[Figure]

**References**

Beven, K. (2006). A manifesto for the equifinality thesis. *Journal of Hydrology*, 320:18–36. 10.1016/j.jhydrol.2005.07.007.

Clark, M. P., Hendrikx, J., Slater, A. G., Kavetski, D., Anderson, B., Cullen, N. J., Kerr, T., Hreinsson, E. O., and Woods, R. A. (2011). Representing spatial variability of snow water equivalent in hydrologic and land-surface models: A review. *Water Resources Research*, 47:1–23. 10.1029/2011WR010745.

Cortés, G., Girotto, M., and Margulis, S. (2016). Snow process estimation over the extratropical Andes using a data assimilation framework integrating MERRA data and Landsat imagery. *Water Resources Research*, 52:2582–2600. 10.1002/2015WR018376.

Cortés, G. and Margulis, S. (2017). Impacts of El Niño and La Niña on interannual snow accumulation in the Andes: Results from a high-resolution 31 year reanalysis. *Geophysical Research Letters*, 44:6859–6867. 10.1002/2017GL073826.

Emerick, A. A. and Reynolds, A. C. (2013). Ensemble smoother with multiple data assimilation. *Computers & Geosciences*, 55:3–15. 10.1016/j.cageo.2012.03.011.

Evensen, G. (2009). *Data Assimilation: The Ensemble Kalman Filter*. Springer-Verlag Berlin Heidelberg. 10.1007/978-3-642-03711-5.

Liston, G. E. (2004). Representing Subgrid Snow Cover Heterogeneities in Regional and Global Models. *Journal of Climate*, 17(6):1381–1397. 10.1175/1520-0442(2004)017<1381:RSSCHI>2.0.CO;2.

Riggs, G. A. and Hall, D. K. (2016). MODIS Snow Products Collection 6 User Guide. Version 1.0.

Slater, A. G., Barrett, A. P., Clark, M. P., Lundquist, J. D., and Raleigh, M. S. (2013). Uncertainty in seasonal snow reconstruction: Relative impacts of model forcing and image availability. *Advances in Water Resources*, 55:165–177. 10.1016/j.advwatres.2012.07.006.

Sturm, M. and Wagner, A. M. (2010). Using repeated patterns in snow distribution modeling: An Arctic Example. *Water Resources Research*, 46:1–15. 10.1029/2010WR009434.

---

## Author Response (AR2)

**Author's response**

We are grateful to the editor and the reviewer for their comments and suggestions to our revised manuscript. In the following author's response we provide a point-by-point reply to all issues raised by the reviewer. The reviewer's comments appear in bold font and our replies in normal font. Excerpts from and changes to the manuscript are quoted in italics. As in the manuscript, $N_t$, $N_o$, $N_e$ and $N_s$ denote the number of time steps, observations, ensemble members and state variables. Page and line numbers refer to positions in the previously revised manuscript. This is followed by a revised version of the manuscript, with changes marked in bold font.

**Ensemble-based assimilation of fractional snow covered area satellite retrievals to estimate the snow distribution at Arctic sites**

**Aalstad et al., 2017, The Cryosphere**

**The revised paper and the responses of the authors are very helpful and clarify the paper. There are a few minor points left that need attention:**

**1) Check the text one more time in depth. Obvious textual mistakes (such as abstract L.4 "a simple simple model", p.11 L9: "the resolution... do\*es\* not...", etc.) give an impression of carelessness.**

The two sentences running from L4 to the middle of L6 in the abstract were merged and corrected to

*"We present an ensemble-based data assimilation framework that estimates the peak subgrid SWE distribution (SSD) at the 1 km scale by assimilating fractional snow covered area (fSCA) satellite retrievals in a simple snow model forced by downscaled reanalysis data."*

In line with the above change, we have also adapted the abstract slightly to better reflect the terminology used later in the paper.

On L13 of the abstract *"Ny Ålesund"* was changed to *"Ny-Ålesund"* to be consistent with the remainder of the text.

The line *"the resolution of the downscaled forcing data do not"* on P11 L9 was changed to *"the resolution of the downscaled forcing data does not"*.

On P11 L19 *"an linear"* was changed to *"a linear"*.

On P12 L22 *"throughout integration of the model for each water year"* was changed to *"throughout the annual integration of the model"*.

The sentence on P14 L7 was changed to

*"Then, we add $N_e$ realizations of Gaussian white noise with a consistent variance to the transformed mean and subsequently apply the inverse transform."*

The sentence starting on P16 L2 was changed to

*"For $N_a = \alpha^{(\ell)} = 1$, the ES scheme, which was used in the probabilistic SWE reconstruction of Durand et al. (2008) and Girotto et al. (2014), is recovered."*

The sentence starting on P16 L26 was changed to

*"Marginal cumulative distributions are recovered through the individual ranking of the ensembles of state vari-*

*ables and perturbation parameters followed by a cumulative summation of the correspondingly sorted weights. These distributions allow for the estimation of quantile values."*

On P18 L26 "subgrid SWE distributions" was changed to "peak subgrid SWE distributions".

On P19 L2 "most likely" was changed to "possibly".

On P19 L4 "an evaluation" was changed to "a direct evaluation".

On P20 L2 "with" was changed to "at".

On P23 L8 "improvement of FI" was changed to "increase in FI".

Removed the units for $\mu$ in Table 6 since all the metrics are dimensionless.

Fixed the use of ':' and ';' in the caption of Figure 8.

**2) It would be helpful to identify all "perturbation parameters" (or estimated parameters) in the descriptive text about the model. It is done now for $Q_0$, it can also be done for $\chi$ and $\alpha_{\mathbf{min}}$.**

We agree. As such, we have added the following sentences

*"$\chi$ is a perturbation parameter (see Table 4) that is updated in the assimilation."*

*"$\alpha_{min}$ is a perturbation parameter (see Table 4) that is updated in the assimilation."*

to P8 L4 and P9 L3 respectively.

**3) Link the observation operator H with the model description in the previous section. H is not linear, yet it is portrayed as a matrix which is not right.**

Since the observed variable (the fSCA) is part of our state, the observation operator is linear. The observation operator becomes a binary matrix $\mathbf{H}$ of dimensions $N_o \times (N_s \times N_t)$ (i.e. with $N_o$ rows and $N_s \times N_t$ columns). This operator simply picks out the ensemble of predicted observations, i.e. the predicted fSCA ensemble at each day where an observation exists, from the ensemble of states as expressed in equation (18). Note that augmenting the state with the observed variable, as we have done, is a standard way of linearizing the observation operator (Evensen, 2003). To clarify, the sentence starting on P14 L21 was changed to:

*"$\mathbf{H}$ is the linear observation operator, a binary $N_o \times (N_s \times N_t)$ matrix, that picks out the predicted fSCA observations from the ensemble of states and $\mathbf{R}$ is the $N_o \times N_o$ diagonal observation error covariance matrix containing the observation error variances (Section 3.2)."*

To highlight that we are describing our specific implementation of the ES-MDA, ES and PBS we have changed the sentence on P14 L12 to:

*"Here, we describe our implementation of three batch smoother schemes: the ES-MDA, the ES and the PBS. The ES-MDA is our focus while the two latter schemes are used for comparison."*

**4) I do not think that it is useful/correct to define the mean SWE depth as a separate state variable in the state vector: the observation operator can use the individual SWE elements to diagnose the mean.**

We agree and have removed the mean SWE depth ($\overline{D}$) from the state. Thus, $\overline{D}$ is removed from the list of state variables on P14 L18 and we have excluded equation (2) from the definition of the model operator on P15 L9.

**5) Why are the obs y not capitalized in the description of the PBS, whereas they are in the ES-MDA? The obs are still a vector of multiple obs in the PBS.**

Following the classical DA notation laid out in Ide et al. (1997), we let bold lower case roman symbols denote vectors and bold upper case roman symbols denote matrices. Thereby, as stated on P14 L19, $\mathbf{y}$ denotes the $N_o \times 1$ observation vector that contains all the available fSCA retrievals during the ablation season. $\mathbf{y}$ is closely related to but not the same as $\mathbf{Y}$ which, as stated on P14 L20, denotes the $N_o \times N_e$ matrix of perturbed observations as defined by equation (19). The PBS, as presented in Margulis et al. (2015), does not use perturbed observations, but instead directly assimilates the unperturbed observation vector $\mathbf{y}$. The ES-MDA, as presented in Emerick and Reynolds (2013), assimilates the perturbed observations $\mathbf{Y}$ in line with the stochastic EnKF (see Burgers et al., 1998) to avoid over-smoothing.

Thank you for your time and consideration as well as all the helpful comments and suggestions,
On behalf of all the co-authors,
Kristoffer Aalstad

**References**

Burgers, G., van Leeuwen, P. J., and Evensen, G. (1998). Analysis scheme in the ensemble Kalman filter. *Monthly Weather Review*, 126(6):1719–1724.

[revised manuscript text omitted]